# New In Situ Aerosol Hyperspectral Optical Measurements over 300-700 nm, Part 1: Spectral Aerosol Extinction (SpEx) Instrument Field Validation during the KORUS-OC cruise

Carolyn E. Jordan[1,2], Ryan M. Stauffer[3], Brian T. Lamb[4], Charles H. Hudgins[2], Kenneth L. Thornhill[2,5], Gregory L. Schuster[2], Richard H. Moore[2], Ewan C. Crosbie[2,5], Edward L. Winstead[2,5], Bruce E. Anderson[2], Robert F. Martin[2], Michael A. Shook[2], Luke D. Ziemba[2], Andreas J. Beyersdorf[2,6], Claire E. Robinson[2,5], Chelsea A. Corr[2,7]and Maria A. Tzortziou[3,4]

[1]National Institute of Aerospace, Hampton, Virginia, United States of America
[2]NASA Langley Research Center, Hampton, Virginia, United States of America
[3]NASA Goddard Space Flight Center, Greenbelt, Maryland, United States of America
[4]City University of New York, New York, New York, United States of America
[5]Science Systems and Applications Inc., Hampton, Virginia, United States of America
[6]California State University, San Bernardino, California, United States of America
[7]Springfield College, Springfield, Massachusetts, United States of America

*Correspondence to*: C. E. Jordan (Carolyn.Jordan@nasa.gov)

**Abstract.** In situ observations of spectrally-resolved aerosol extinction coefficients (300-700 nm at ~0.8 nm resolution) from the May-June 2016 Korea U.S. – Ocean Color (KORUS-OC) oceanographic field campaign are reported. Measurements were made with the custom-built Spectral Aerosol Extinction (SpEx) instrument that previously has been characterized only using laboratory-generated aerosols of known size and composition. Here, the performance of SpEx under realistic operating conditions in the field was assessed by comparison to extinction coefficients derived from commercial instruments that measured scattering and filter-based absorption coefficients at three discrete visible wavelengths. Good agreement was found between these two sets of extinction coefficients with slopes near unity for all 3 wavelengths within the SpEx measurement error ($\pm 5$ Mm$^{-1}$). The meteorological conditions encountered during the cruise fostered diverse ambient aerosol populations with varying sizes and composition at concentrations spanning two orders of magnitude. The sampling inlet had a 50% size cut of 1.3 µm diameter particles such that the in situ aerosol sampling suite deployed aboard ship measured fine mode aerosols only. The extensive hyperspectral extinction data set acquired revealed that nearly all measured spectra exhibited curvature in logarithmic space, such that Ångström exponent ($\alpha$) power law fits could lead to large errors compared to measured values. This problem was particularly acute for $\alpha$ values calculated over only visible wavelengths, then extrapolated to the UV, highlighting the need for measurements in this wavelength range. Second-order polynomial fits to the logarithmically-transformed data provided a much better fit to the measured spectra than the linear fits of power laws. Building on previous studies that used total column AOD observations to examine the information content of spectral curvature, the relationship between $\alpha$ and the second order polynomial fit coefficients ($a_1$ and $a_2$) was found to depend on the wavelength range of the spectral measurement such that any given $\alpha$ maps into a line in ($a_1, a_2$) coefficient space with a slope of $-2LN(\lambda_{ch})$, where $\lambda_{ch}$ is defined as the single wavelength that characterizes the wavelength range of the measured spectrum (i.e., the "characteristic wavelength"). Since the curvature coefficient values depend on $\lambda_{ch}$, it must be taken into account when comparing values from spectra obtained from measurement techniques with different $\lambda_{ch}$. Previously published work has shown that different bimodal size distributions of aerosols can exhibit the same $\alpha$, yet have differing spectral curvature with different ($a_1, a_2$). This implies that ($a_1, a_2$) contain more information about size distributions than $\alpha$ alone. Aerosol size distributions were not measured during KORUS-OC and the data reported here were limited to the fine fraction, but the ($a_1, a_2$) maps obtained from the SpEx data set are consistent with the expectation that ($a_1, a_2$) may contain more information than $\alpha$, a result that will be explored further with future SpEx and size distribution data sets.

## 1 Introduction

Significant natural variability in the size and composition of atmospheric aerosols introduces uncertainty into the representation of their optical properties and radiative impacts in models and satellite retrievals. While numerous airborne and surface-based observations over past decades have placed important constraints on these relationships, measurements have been typically limited to only one or a few wavelengths of light, which are extrapolated across the ultraviolet (UV)-visible-infrared (IR) spectrum by assuming a power law relationship (e.g., the well-known Ångström exponent). However, there is long standing evidence that extinction and aerosol optical depth spectra in the ambient atmosphere exhibit curvature that is not fully captured by a power law (e.g., King and Byrne, 1976; King et al., 1978; Kaufman, 1993; Reid et al., 1999; Eck et al., 1999, 2001a, b, 2003a, b; O'Neill et al., 2001; Schuster et al., 2006; Kaskaoutis et al., 2010, 2011; Rao and Niranjan, 2012). This wavelength-dependence of the aerosol extinction is thought to be driven primarily by the particle size distribution with only a minor contribution from the compositionally dependent aerosol absorption (Eck et al., 2001b; Schuster et al., 2006). However, these relationships remain largely qualitative and stem from columnar remote sensing measurements and Mie theory calculations. This motivates the incorporation of recently

developed, advanced instruments for measuring in situ, hyperspectral aerosol extinction into field campaigns that study the ambient atmosphere.

It is important to note that the power law assumption is empirical. Used to describe the interaction between light and aerosols (e.g. Ångström, 1989; Moosmüller and Chakrabarty, 2011),

$$p(\lambda) = \beta \lambda^{-\alpha} \tag{1}$$

p may represent various optical parameters of interest (e.g., scattering, absorption, extinction), $\lambda$ is the wavelength of light ($\mu$m), $\beta$ is p at 1 $\mu$m, and $\alpha$ is the Ångström exponent. One advantage of this representation is that the derivative of this relationship produces a line in logarithmic space with a wavelength-independent slope,

$$\alpha = -\frac{d\,LN(p(\lambda))}{d\,LN(\lambda)} \tag{2}$$

Hence, from any 2 wavelengths, the Ångström exponent for the entire spectrum can be determined via

$$\alpha = -\frac{LN(p(\lambda_2)/p(\lambda_1))}{LN(\lambda_2/\lambda_1)} \tag{3}$$

The simplicity of this representation has led to its widespread use in the aerosol and remote sensing communities due to its utility in calculating p($\lambda$) at wavelengths for which there isn't a direct measurement. $\alpha$ is sensitive to the measurement error in p($\lambda$), such that if multiple wavelengths are available, it is preferable to perform a linear fit using Eq. (2) than a paired wavelength calculation as in Eq. (3). If the optical property of interest is fully described by a power law, then further spectral detail beyond several wavelengths is superfluous. However, if $\alpha$ is not, in fact, wavelength-independent for ambient aerosols, then a hyperspectal measurement (or sufficient wavelength sampling spanning the full wavelength range of interest) is required to capture the actual wavelength dependence.

Previous studies have considered various aerosol properties that may influence the spectral shape (and/or variation of the Ångström exponents calculated from different pairs of wavelengths) of ambient aerosol optical depth (AOD) such as the volume mean (or effective) radius of accumulation mode aerosols, the geometric standard deviation (i.e., width) of the accumulation mode, the volume fraction of the fine mode relative to the total aerosol population, and the absorption characteristics (i.e., composition) of the aerosol population (e.g., Reid et al., 1999; Eck et al., 1999, 2001b; Schuster et al., 2006). The relationships identified to date remain qualitative rather than quantitative and have been based on total atmospheric column measurements from remote sensors and on Mie calculations. Recently, the retrieval algorithm developed by O'Neill et al. (2001, 2003, 2008) to distinguish fine mode from coarse mode AOD components using 2nd order spectral fits has been applied to in situ extinction components based on scattering and absorption measurements made at a few visible wavelengths and evaluated using several sets of field data (Kaku et al., 2014, and references therein). The Spectral Aerosol Extinction (SpEx) instrument (Jordan et al., 2015) provides a new measurement approach that combines the advantages of a broad spectral range (typically limited to remote sensing techniques) with an in situ measurement capability (that allows for direct comparison to other in situ measurements of ambient aerosol microphysical and chemical properties). Combining this hyperspectral in situ measurement capability with retrieval techniques as in Kaku et al. (2014) also provides a new tool to fine tune remote sensing retrievals.

Although SpEx measures a narrower wavelength range (300 - 700 nm) than some remote sensors (e.g., ground-based 9 band sun-sky radiometers AERONET (340 - 1640 nm, Giles et al., 2019) and airborne hyperspectral sun-sky radiometer 4STAR (350 - 1750

nm, 2 - 3 nm resolution, LeBlanc et al., 2020)) and similar ranges to others (e.g., hyperspectral polarimeters SPEX airborne (400 - 800 nm, 2 - 3 nm resolution in intensity, Smit et al., 2019; Fu et al., 2020) and SPEXone (385 - 770 nm, 2 nm resolution in intensity, Hasekamp et al., 2019) planned for the upcoming PACE mission (Werdell, et al., 2019; Remer et al., 2019a, 2019b)), it measures deeper into the UV range and has finer spectral resolution, ~0.8 nm.  Similar in situ instruments, such as broadband cavity-enhanced

spectroscopy (BBCES) instruments (e.g., Washenfelder et al., 2013, 2015; Bluvshtein et al., 2016, 2017; He et al., 2018), do not provide as broad a UV-visible range as SpEx.  Hence, SpEx is particularly suited to examine spectral details for ambient aerosols over its measurement range and to relate those spectral details to simultaneous in situ measurements of ambient aerosol microphysics and composition.

Previous work described SpEx using a series of laboratory tests that included a variety of non-absorbing and absorbing aerosols to characterize the instrument (Jordan et al., 2015).  However, the purpose of SpEx is to measure atmospheric aerosols in the ambient environment.  Hence, this manuscript is the first to offer details on the instrument performance in the field.  The data presented here were obtained during the Korea United States - Ocean Color (KORUS-OC) oceanographic field campaign conducted around the Korean peninsula under the leadership of the Korean Institute of Ocean Science and Technology (KIOST) and the U.S. National

Aeronautics and Space Administration (NASA).  The KORUS-OC cruise was affiliated with the airborne KORUS - Air Quality (KORUS-AQ) campaign and the KOrean Coastal water Ocean and Atmosphere (KOCOA) field campaign.  These joint missions were conducted to study South Korean air quality and ocean color within the field of regard of South Korea's Geostationary Ocean Color Imager (GOCI) that provided hourly ocean color and aerosol optical depth (AOD) measurements.  Although the primary scientific objectives of the KORUS-OC cruise focused on ocean color measurements (both in situ and remotely sensed), there were

also objectives to address atmospheric correction requirements and to explore interdisciplinary science questions (e.g., Tzortziou et al., 2018; Thompson et al., 2019).  The joint S. Korean and U.S. based science teams sailed aboard the KIOST research vessel *R/V Onnuri* (Fig. S1).  Details on the scientific objectives of these field campaigns are provided in two white papers: Al Saadi et al. (2015) and US-Korean Steering Group (2015).  An overview of the findings from KORUS-AQ is provided by Crawford et al. (2020) and references therein.


Two commercial instruments (AirPhoton's integrating nephelometer, IN101, and Brechtel's Tricolor Absorption Photometer, TAP) were also deployed to measure in situ aerosols at three visible wavelengths providing scattering coefficients and absorption coefficients, respectively.  These measurements are presented in detail here and used to evaluate the performance of SpEx.  Filter samples were also collected and analyzed in the laboratory for spectral absorption and chemical composition as discussed in Part

2 of this work.  Those data provide additional context for the measurements described here in Part 1.  Further, Part 2 presents an assessment of the applicability of a methodology routinely used by the ocean color community to measure hyperspectral particle absorption for use with ambient atmospheric aerosol samples.  Hence, Part 2 includes further discussion of the ability of power laws to fully represent the observed hyperspectral variability of in situ aerosol optical properties as measured during KORUS-OC.

## 2 Methods

### 2.1 Ship deployment

The measurements reported here from 20:25 May 21 through 09:00 June 4, 2016 Korean Standard Time (KST, KST = UTC + 9) were made outside of South Korea's territorial seas (> 12 nautical miles, 22.2 km, from the coast, Fig. 1, top left panel).  The

instrument suite (Fig. S2) was deployed above the bridge strapped to the starboard rail in a custom-built box designed to keep the instruments dry yet ventilated to prevent overheating (Fig. S1). The pumps were located in a separate box a few meters away tied

to the stern rail (Fig. S1) to limit thermal and mechanical interference with the measurements. Measurements were made at ambient temperature (T), pressure (P), and relative humidity (RH), (i.e., the aerosols were not dried prior to measurement) with the exception of the TAP instrument which is internally heated and kept at a constant 35 °C. T, P, and RH were measured both aboard ship and within the sampling system. Although the sampling box was ventilated, it was not climate controlled. On the top deck it was in direct sunlight during the day and even at night various components in the system kept the interior warmer and drier than

ambient air. These conditions led to diurnal variability in the difference between sampling and ambient conditions, such that the sampling T was ~ 6 °C warmer at night increasing to as much as ~15 °C warmer in mid-afternoon. Similarly, ambient RH ranged from 55 - 98% RH throughout the cruise, while the sampling RH ranged from ~ 30 - 70%. For the purposes of this work (both Parts 1 and 2), the comparisons made are all within this sampling system with T and RH consistent across all measurements, except for TAP. Since non-volatile black carbon (BC) is expected to dominate the absorption measurement, this difference is expected

to have a limited impact on the results. The primary objective in this work is to evaluate the performance of SpEx by direct comparisons to the data from the two commercial instruments in the measurement suite. Hence, we did not perform any corrections to either ambient T and RH or to standard temperature and pressure (STP) prior to comparing these data. There are no comparisons in this pair of manuscripts to other data sets, so such corrections are not necessary for this study.

The available berths aboard ship constrained the number of personnel available to operate the sampling system, so it was necessary to configure the system to operate nearly autonomously. To ensure that water did not enter the sampling line, a tall stainless steel sampling mast (19.05 mm (3/4") inner diameter tubing) was used to minimize the chance of sea spray entering the line from below. Further, the inlet was attached to the vertical sampling mast with a curved section of stainless steel tubing such that the inlet was downward facing to prevent any potential precipitation from entering the line from above as well (Fig. S1). This configuration

worked as intended with the inlet approximately 10 m above the sea surface. However, the fast flow rate required for the SpEx measurement (approximately 70 lpm) through the curved inlet resulted in the removal of coarse aerosol with a 50% size cut of ~1.3 µm diameter (Fig. S3). Hence, all of the measurements here (sampling from the same inlet, Fig. S2) reflect only the fine fraction aerosol in the marine boundary layer (MBL).

While the SpEx and commercial instruments were run mostly uninterrupted, colleagues from other groups aboard ship carried out the routine filter changes needed throughout the cruise. They also ensured that the system ran properly and were there to handle problems. The methodology and results of the filter sampling will be presented in the companion paper (Part 2, Jordan et al., 2020b).

## 2.2 Aerosol scattering and absorption coefficient measurements and derived parameters

The IN101 and TAP instruments provide data at a higher temporal resolution than SpEx and were deployed with two objectives: 1) to identify and flag incidents of ship exhaust ("plume") contamination of the data set, and 2) to evaluate the new spectral measurements (both from SpEx and the filters). The TAP (model 2901, Brechtel, Hayward, CA) measures absorption coefficients ($\sigma_{abs}$) at 467, 528, and 652 nm with 1 s resolution and the IN101 (AirPhoton, Baltimore, MD) measures scattering coefficients ($\sigma_{scat}$) at 450, 532, and 632 nm with ~10 s resolution.


To identify ship plume interceptions, an initial examination of the IN101 $\sigma_{scat}$ and TAP $\sigma_{abs}$ data was performed. One-minute averages were calculated in order to calculate the single scattering albedo ($\omega = \sigma_{scat}/(\sigma_{scat} + \sigma_{abs})$). The $\sigma_{scat}$ data was corrected using the submicron factors in Anderson and Ogren (1998). These corrections are needed to resolve truncation errors in the forward scattering (the 7° to 0° range missing from the measurement) and non-Lambertian errors that arise from the distribution of light by the opal glass diffusor. Due to limited personnel, calibrations were not performed during the cruise. Pre- and post-cruise calibrations using pure $CO_2$ were performed in the laboratory to correct the final archived data and indicate the instrument performance was stable throughout the measurement period. The $\sigma_{abs}$ data was corrected for scattering as recommended in Ogren et al. (2017).

Interception of the ship plume was first identified using one-minute averages of the gas-phase $NO_2$ and $O_3$ measurements (Thompson et al., 2019). The inlets for the gas-phase instruments were near, but not co-located with the aerosol inlet, so the one-minute averages of $\omega$, $\sigma_{abs}$, and $\sigma_{scat}$ were used to further assess evident ship plume interceptions in the aerosol data set. The $\sigma_{abs}$ data were particularly sensitive to such interceptions exhibiting dramatic brief spikes in the otherwise smoothly varying time series (the interested reader can compare $\sigma_{abs}$ in Fig. S4 that shows all of the data in the time series including ship plume interceptions to $\sigma_{abs}$ in the top panel of Fig. 2 where the plume interceptions have been removed). These were easily identifiable and removed manually. The inlets for the in situ instruments were towards the starboard bow (gas-phase on a lower deck, a few meters aft of the aerosol inlet) opposite of the ship stack pointed toward the port side stern. While underway, ship plumes did not affect the measurements, but they were encountered occasionally when on station or, as was more frequently the case, when getting underway from a stationary sampling site. The archived plume flags (= 0 for ambient air, 1 for ship plume interceptions) may be used to either remove data points affected by ship contamination or to assess ship plume contamination of filter samples (see Part 2).

In addition to the 1 min averaged $\sigma_{scat}$ and $\sigma_{abs}$ data reported in the KORUS-AQ data archive, averages were also calculated for each 30 s SpEx sampling interval and corrected in the same manner as the 1 min data set. The IN101 and TAP instruments measure at different wavelengths, so in order to calculate $\sigma_{ext}$ ($= \sigma_{scat} + \sigma_{abs}$), the TAP data were adjusted to the IN101 wavelengths using Eq. (3). For clarity, these extinction coefficients are denoted NT (for Nephelometer + TAP) $\sigma_{ext}$ in comparison to the measured SpEx $\sigma_{ext}$ at those wavelengths (450, 532, and 632 nm). Similarly, a second set of plume flags was created for the SpEx sampling interval (i.e., based on plume interceptions for 30 s intervals approximately every 4 minutes, see Sect. 2.3).

**2.3 Spectrally-resolved aerosol extinction coefficient measurements**

Developed from a prototype described in Chartier and Greenslade (2012) SpEx is described in detail in Jordan et al. (2015). SpEx is a White-type optical cell (White, 1942) with a 39.4 m path length and a 17 l internal volume. A flush time of 90 s completely exchanges the air in the cell (3 times the volume) for flow rates > 34 lpm, here, the flow rate was ~70 lpm. The optical cell is coupled via fiber optics to a UV5000 system (Cerex Monitoring Solutions, LLC, Atlanta, GA) with a 150 W xenon lamp source (Cerex P/N CRX-X150W), integrated with an Ocean Optics, Inc. (Dunedin, FL) QE65Pro 16-bit spectrometer. Sampling for 30 s using an integration time of ~20-50 ms optimizes the signal to noise ratio for each measured intensity spectrum (Jordan et al., 2015). An automated valve system controls the 4-minute sampling cycle by switching the flow between the filtered line (ambient air without aerosols) and an unfiltered line (ambient air with aerosols): 90 s flush, 30 s sample without aerosols, 90 s flush, 30 s sample with aerosols, repeat.

The intensity spectrum is measured for the sample without aerosols for a reference ($I_0(\lambda)$) and for the sample with aerosols ($I(\lambda)$) from which the extinction spectra ($\sigma_{ext}(\lambda)$) are calculated using the extinction law (Eq. (4)),

$$\sigma_{ext}(\lambda) = \frac{-LN(I(\lambda)/I_0(\lambda))}{L} \tag{4}$$

where $\lambda$ is the wavelength of light and L is the optical path length (here, in units of Mm). Hence, extinction spectra are acquired every 4 minutes. Reference spectra before and after each sample spectrum are averaged together to account for drift in the intensity between measurements in the calculation of $\sigma_{ext}$. A particular strength of this measurement is it explicitly accounts for extinction arising from gas-phase constituents via the reference spectra. No calibrations are needed to obtain aerosol $\sigma_{ext}$. Laboratory tests have shown the measurement error over the full spectral range of the measurements (300-700 nm) to be about $\pm$ 5 Mm$^{-1}$ (Jordan et al., 2015).

Several modifications have been made to the instrument since the laboratory studies reported in Jordan et al. (2015). These include an automated switching system between valves that control flow between filtered and unfiltered air and new custom control software that allows for continuous unattended operation. Further, the metal-jacketed optical fibers used previously were replaced with bare optical fibers securely packed in foam to reduce noise when deployed on mobile platforms. This last change resulted in greatly improved data quality with far fewer spectra disrupted due to mechanical disturbance under field conditions. Nonetheless, some spectra had to be rejected from the final data set due to features that provided clear evidence of disturbance. Typically, this occurred around the times of the filter changes when the lid of the box housing the sampling system needed to be raised to change filters (flagged using the plume flag field, value set to 2, see Fig. S4). Sometimes the filters were changed during flush times, leaving measured spectra unaffected, while at other times, disturbed spectra persisted for some period of time around the filter change. This suggests other vibrations arising from work involving nearby instruments or elsewhere on the ship may have contributed to the noisy rejected spectra. More work is required to further reduce the susceptibility of SpEx to sources of vibrational noise; nonetheless < 8% of the 4255 measured spectra were rejected from the data set overall. Note, ship plume interception (affecting < 6% of the measured spectra) often resulted in distorted spectra likely due to rapid changes in both the aerosol population and the gas-phase concentrations used to establish the reference spectrum for the calculation of aerosol extinction as in Eq. (4). Hence, only spectra obtained when the plume flag = 0 should be used for analysis of ambient conditions.

One consequence of unattended operation is the intensity drift sometimes resulted in saturated pixels around 332 nm and 467 nm due to peaks at those wavelengths in the lamp spectrum (Fig. S5). Saturation can be prevented either by adjusting the integration time of the spectrometer or by realigning the optics. However, unattended operation necessitated filtering the spectra at those two wavelengths when saturation occurred. The 16-bit spectrometer has a maximum intensity count of 65,536 ($2^{16}$), however, saturation effects became apparent before reaching this maximum. Tests showed that a threshold of 63,000 counts was a suitable limit for removing saturation effects from the spectrum. Filtering the 467 nm channel and the two adjacent pixels on either side of that channel removed the saturation distortions completely. The saturation effects at 332 nm were filtered in the same way, however, there appeared to be a minor shoulder effect that extended over a broader wavelength range in the UV. The effect was minimal (a few percent of the measured value) and well within the measurement error. However, caution is recommended against over-interpreting the shape of a spectrum in the vicinity of 332 nm when the saturation flag (Sat332Flag) for this channel is set to 1. Saturation at 332 nm affected 16% of the total 4255 measured spectra.

# 3 Results

## 3.1 Overview of $\sigma_{scat}$ and $\sigma_{abs}$ during KORUS-OC

Four distinct synoptic meteorological regimes have been described in detail for the KORUS-AQ period (Peterson et al., 2019). The KORUS-OC cruise departed the peninsula (Fig. 1, top left panel) sailing to the East Sea (Sea of Japan) near the end of the 2nd of these periods, the Stagnant period, characterized by limited transport and enhanced photochemical production of secondary organic aerosols (Kim et al., 2018; Nault et al., 2018; Choi et al., 2019; Jordan et al., 2020a). This regime started breaking down on May 23rd, followed by a precipitation event on the 24th that reduced ambient aerosols to low concentrations as reflected by low (tens of Mm$^{-1}$) $\sigma_{scat}$ (Fig. 2, top panel).

From May 25th through the 31st the Transport/Haze period was characterized by air mass transport (from the west/northwest carrying pollutants from China), overcast hazy conditions, and rapid local South Korean secondary production of inorganic aerosols resulting in the largest concentrations of the PM$_{2.5}$ fraction of aerosols (i.e., particulate matter with diameters $\leq$ 2.5 µm) observed during the KORUS-AQ campaign (Peterson et al., 2019; Eck et al., 2020; Jordan et al., 2020a). The greatest $\sigma_{scat}$ values (hundreds of Mm$^{-1}$) observed aboard the *R/V Onnuri* were found during the first half of the Transport/ Haze period while the ship was downwind of the Korean peninsula in the East Sea (Fig. 1, top panels, and Fig. 2, top panel). Lower $\sigma_{scat}$ values were observed during this period when the ship was upwind of the peninsula following its transit to the West Sea (Yellow Sea).

The final of the four synoptic periods, the Blocking period, followed a frontal passage that ended the previous Transport/Haze period (Peterson et al., 2019). This final frontal passage swept in cleaner air from the north leading to a rapid decrease in aerosol concentrations reflected in the reduced $\sigma_{scat}$ observed aboard ship (Fig. 2). The Blocking period was then characterized by limited transport and occasional brief stagnant periods due to adjacent high and low pressure systems with the high poleward of the low (called a Rex Block). Under these conditions local sources dominated pollutants, but aerosols did not accumulate to large concentrations (Peterson et al., 2019; Jordan et al., 2020a). This led to strikingly low $\sigma_{scat}$ values (tens to < 10 Mm$^{-1}$, Fig. 2), even lower than those observed during the precipitation event on May 24th.

Throughout the cruise $\sigma_{abs}$ values were typically an order of magnitude smaller than $\sigma_{scat}$ (Fig. 2, top panel) with peak values generally found at the same locations and times as peak $\sigma_{scat}$ (Fig. 1). Hence, while the temporal variability of $\sigma_{abs}$ largely followed $\sigma_{scat}$, the range in the magnitude of $\sigma_{abs}$ (0.36 - 18.07 Mm$^{-1}$) at 532 nm was far less than that of $\sigma_{scat}$ (2.8 - 332.2 Mm$^{-1}$). This led to the finding that single scattering albedo values ($\omega(\lambda) = \sigma_{scat}(\lambda) / \sigma_{ext}(\lambda)$) were driven by the change in scattering not absorption. Typically, $\omega$ was > 0.9 (Fig. 1, bottom right panel, and Fig. 2, bottom panel). The excursions below this value observed on May 23rd and 24th occurred when the reduction in $\sigma_{scat}$ exceeded that observed in $\sigma_{abs}$.

The most extreme low values of $\omega$ (~ 0.7 for 532 nm, Figs. 1 and 2), however, occurred during the Blocking period, when the temporal evolution of $\sigma_{abs}$ did not closely follow $\sigma_{scat}$. At that time, the scattering Ångström exponents ($\alpha_{scat}$) increased to values > 3, while absorption Ångström exponents ($\alpha_{abs}$) were ~ 1 (Fig. 2, middle panel). These data indicate that the aerosol population was dominated by small particles, likely black carbon (BC). The temporal evolution of carbon monoxide (CO, Fig. 2, bottom panel) and $\sigma_{scat}$ (Fig. 2, top panel) were strikingly similar from 04:00 June 1st through 20:00 June 2nd, exhibiting an r$^2$ = 0.841 for a linear regression over that time. This is in contrast to the Blocking period as a whole (r$^2$ = 0.514) or the rest of the campaign,

excluding the Blocking period ($r^2 = 0.623$). The good correlation with CO, the BC signature in $\alpha_{abs}$, the small particle population indicated by $\alpha_{scat}$, and the limited transport of the Blocking period, together suggest that the low $\omega$ values arose from local ship emissions either from commercial or fishing vessels or both. Note, the inference that aerosols during the Blocking period were from local ship emissions refers to the regional ambient environment and should not be confused with ship plume contamination from the *R/V Onnuri* (see Section 2.2 for the criteria used to remove ship stack contamination from the data set).

Aside from the Blocking period, $\alpha_{scat}$ typically ranged from ~ 1.5 - 2, with evident wavelength dependence in the $\alpha_{scat}$ wavelength pairs (Fig. 2, middle panel). Little difference was found among the wavelength pairs in the $\alpha_{abs}$ values that typically ranged from ~0.5 - 1 throughout the cruise.

### 3.2 Comparison of SpEx data to $\sigma_{ext}$ calculated from $\sigma_{scat} + \sigma_{abs}$

$\sigma_{ext}$ calculated from the $\sigma_{scat} + \sigma_{abs}$ data (denoted NT $\sigma_{ext}$ to distinguish it from SpEx $\sigma_{ext}$) averaged over the SpEx sampling intervals (see Sect. 2.2) were compared to the 450, 532, and 632 nm channels of SpEx (Fig. 3). Excellent agreement was found with slopes of $1.020 \pm 0.002$, $0.998 \pm 0.003$, and $1.057 \pm 0.004$ for all of the data in each of the three channels (gray plus markers in all panels of Fig. 3), respectively. Three intervals were used to look at mean comparisons: 15 min (light colored circles), 30 min (dark colored triangles), and 60 min (black diamonds). The slopes, intercepts, and $r^2$ values for all of the fits are shown in Fig. 3. These fits were performed using data limited to the valid measurement range (i.e., above the lower limit of detection, LLOD). Tests indicated that a limit of twice the measurement uncertainty provided a suitable LLOD (10 Mm$^{-1}$).

The SpEx data were more variable than the NT $\sigma_{ext}$ as is evident in time series comparisons throughout the cruise (Figs. 4, S6, and S7, top panel). This is partly attributable to the differing noise characteristics of the measurement techniques, but it also arises from differences in sampling intervals where the standard error of the means reduces the variability by the square root of the number of samples in the mean. The NT $\sigma_{ext}$ represent 30 s means calculated from ~10 s $\sigma_{scat}$ and 1 s $\sigma_{abs}$ measurements for each SpEx $\sigma_{ext}$ spectrum. Averaging SpEx $\sigma_{ext}$ reduces the variability (Figs. 4, S6, and S7, middle panel) according to the standard error of the means. However, not all of the variability evident in Fig. 4 is noise. Limiting the data range in the time series to 2 days (Fig. 5) illustrates that the native resolution of SpEx captured rapid changes that were also evident in the NT data. For example, consider the double peak feature that occurred around 03:00 on May 25$^{th}$ (top panel, Fig. 5). This temporal variation is lost even in the 15 min average (bottom panel, Fig. 5).

For clarity, the standard deviations for the means are shown separately (Figs. 4, S6 and S7, bottom panels). Most of the standard deviations of the means in these channels were consistent with the measurement error of about $\pm 5$ Mm$^{-1}$ with larger values arising from changes in the ambient conditions. Hence, the standard deviations of the 60 min means describe an upper envelope compared to the 15 and 30 min means as changing ambient conditions led to greater variability in the means. Together, the results shown in Figs. 3-5 show that quantitative data were obtained under field conditions with SpEx at its native sampling resolution.

### 3.3 Evaluating extinction Ångström exponents ($\alpha_{ext}$) with hyperspectral extinction data

With the spectral data available from SpEx, it was possible to test the power law behavior of the individual and mean spectra by fitting a line to each spectrum in logarithmic space per Eq. (2), where the Ångström exponent, $\alpha_{ext}$ is the negative value of the slope. As discussed in the introduction, if a power law described the relationship well, $\alpha_{ext}$ should be invariant with wavelength.

Yet clear separation was found in $\alpha_{ext}$ from linear fits to all of the individual spectra (All Data, Fig. 6, left panel) depending on the wavelength range used for the fit. The $\alpha_{ext}$ from fits to the full range (300-700 nm), and two partial ranges (450-532 nm) and (532-632 nm) deviate from the 1:1 line expected when plotted vs. $\alpha_{ext}$ from fits to the 450-632 nm range. This result was also found for $\alpha_{ext}$ determined from all of the mean spectra sets as well (e.g., those from the 30 min mean spectra, right panel Fig. 6).


Most of the SpEx spectra measured during the cruise exhibited curvature over the 300-700 nm range in logarithmic space. Note, in order to compare results here to previously published work (Sect. 4), for the remainder of this section fits will be shown using wavelength in units of μm. This change in units does not alter the values of $\alpha_{ext}$ (which are invariant with wavelength and hence, the choice of units does not matter), but it does change the other coefficients from the mathematical fits to the spectra as will be

discussed further in Sect. 4. An example of the observed curvature is provided in Fig. 7. The measured spectrum shown (red curve, Fig. 7, all panels) is curved such that residuals (the difference between the measured spectrum and the mathematical fit, blue curves) from the linear fits (black lines, left panels) are also curved. If a particular mathematical function (here, a power law) provides a good fit to the data, the residuals should be randomly distributed around zero. If there is a trend (the curvature evident here), then other functions should be considered to see if a better fit may be obtained. The curvature in the residuals is evident not

only across the full wavelength range (Fig. 7, bottom left panel), but for subsets of that range as well (e.g., the 0.45 - 0.632 μm fit, Fig. 7, top left panel). Note, the log scale used to plot the spectrum (red curves) in Fig. 7 along with the relatively small extinctions at long wavelengths exaggerates the appearance of noise at those wavelengths (i.e., ± 5 Mm$^{-1}$ at the red end where $\sigma_{ext} \sim 20$ Mm$^{-1}$, LN(20) ~ 3, is more obvious than in the UV where $\sigma_{ext} \sim 150$ Mm$^{-1}$, LN(150) ~ 5). Nonetheless, the intensity of the xenon lamp decreases from 600 to 700 nm (Fig. S5; in Fig. 7, LN(0.6 μm) ~ -0.51, LN(0.7 μm) ~ -0.36) such that smaller values of $I_0$ combined

with the small differences between I and $I_0$ in this wavelength range lead to slightly greater uncertainty in $\sigma_{ext}$ (Eq. (4)).

The wavelength dependent curvature can lead to large errors if used to extrapolate to wavelengths beyond the measured range of values. For example, using $\alpha_{ext}$ found from the 0.45-0.632 fit to extrapolate to 0.3 μm (Fig. 7, middle left panel) leads to a value of LN($\sigma_{ext}$) = 5.7099, while the measured value was 5.0839, i.e., an extrapolated extinction of 302 Mm$^{-1}$, 87% larger than the

measured extinction of 161 Mm$^{-1}$. This wavelength is the most extreme, but the positive artifact is present throughout the UV range: with the extrapolated extinction 73% and 29% greater at 0.315 and 0.365 μm, respectively. At 0.45 μm (LN(0.45 μm) = -0.8, Fig. 7), the lower end of the fit range of values, the fit extinction was 6% greater than the measured value, while at 0.532 μm (the middle of the fit range) it was 3% less.

Previous work has shown that 2$^{nd}$ order polynomials in logarithmic space can provide a better fit to ambient aerosol optical depth (AOD, aerosol extinction integrated over an atmospheric column) spectra than power law fits (e.g., Eck et al., 1999, 2001a, b, 2003a, b; Schuster et al., 2006; Kaskaoutis et al., 2010, 2011). In the example shown in Fig. 7, it is clear that a 2$^{nd}$ order polynomial fit (black curves, right panels) reduces the trends in the residuals both over the full wavelength range (bottom right) and over the 0.45-0.632 μm subset of that range (top right) from that obtained from the linear fits. The improved fit provided by a 2$^{nd}$ order

polynomial for all of the measured spectra is shown using histograms of the residuals at 6 wavelengths across the measured wavelength range (Fig. 8). For both the linear fits (black bars) and the 2$^{nd}$ order polynomial fits (red bars), the range of values for residuals at each wavelength is divided into 20 bins. The narrower bins for the 2$^{nd}$ order polynomial fits reflect the smaller range in residual values compared to those from the linear fits. The best agreement between the two sets of residuals shown in Fig. 8 is found at 0.532 μm, where the linear fit residuals are distributed around zero. At longer and shorter wavelengths, however, the

linear residuals tend to be either positive or negative at any given wavelength, while the 2$^{nd}$ order polynomial fit residuals are

centered around zero across all wavelengths. These results confirm that $2^{nd}$ order polynomials provide a better fit to the data than linear fits.

Unfortunately, just as extrapolating linear fits beyond the measurement range is problematic, the same is also true for the $2^{nd}$ order polynomial fits (Fig. 7, middle right panel). In this case, at 0.45 and 0.532 μm (within the fit range) the fit agrees with the measured extinctions within 1% (~85 and 56 Mm$^{-1}$, respectively). However, at increasingly shorter wavelengths the fit diverges from the measured spectrum with the fit values 15%, 28%, and 34% too small at 0.365, 0.315, and 0.3 μm, respectively. The divergence of either fit from the measured spectrum when extrapolating beyond the fit wavelength range (Fig. 7, middle panels) highlights the need for measurements across a broad spectral range in order to minimize the need for extrapolation.

## 4 Discussion

The motivation to fit the SpEx spectra with a $2^{nd}$ order polynomial came primarily from the work of Schuster et al. (2006) in which both Mie calculations and AERONET data were used to explore the additional information that spectral curvature may provide. A comparison of the coefficients obtained from SpEx to the fine fraction subset of aerosols reported in Schuster et al. (2006) revealed two key differences between the data sets. First, the $a_1$ and $a_2$ coefficients spanned a wider range of values than those obtained in the prior work (Fig. 9, top left panel). Second, Schuster et al. (2006) reported an empirical approximation such that $\alpha_{ext}$ was approximately equal to $a_2-a_1$. This approximation clearly does not hold for the values obtained from this data set (Fig. 9, top left panel). These differences can be understood as follows.

The two expressions used here to fit the relationship between $\sigma_{ext}$ and $\lambda$ are related by their negative derivative, defined as $\alpha$ in Eq. (2). That is, the derivative of the linear fit (y=a+bx; dy/dx = b) equals the derivative of the $2^{nd}$ order polynomial fit (y = $a_0$ +$a_1$x +$a_2$x$^2$; dy/dx = $a_1$ +$2a_2$x) such that

$$\alpha_{ext} = -b = -(a_1 + 2a_2(LN(\lambda)) \tag{5}$$

Note, the derivative of Eq. (5), $\alpha_{ext}'$ = - d $\alpha_{ext}$/ d LN($\lambda$) = -$2a_2$ defines the curvature of the extinction spectra (Eck et al., 1999). For any given spectrum, there is one wavelength at which the linear and $2^{nd}$ order polynomial fits yield equivalent results in Eq. (5). This must not be confused with every wavelength measured in the spectrum, so we will refer to this one wavelength as the characteristic wavelength of the measurement range, $\lambda_{ch}$, from here on. It can be calculated for each measured spectrum from the two sets of fit coefficients for that spectrum. That is, rewriting Eq. (5) in terms of $\lambda_{ch}$ the characteristic wavelength of the measured spectrum may be calculated from $\lambda_{ch}$ = e$^{\wedge}$(($\alpha_{ext}$ + $a_1$) / -$2a_2$). For the SpEx data set, $\lambda_{ch}$ was found to range from 0.36 - 0.46 μm. In contrast, the empirical fit of $\alpha_{ext}$ = $a_2-a_1$ implies $\lambda_{ch}$ ~ 0.61, i.e., LN(0.61) ~ -0.5. The dependence of Eq. (5) on the characteristic wavelength, results in spectra sets with differing $\lambda_{ch}$ exhibiting different mapping between $\alpha_{ext}$ and ($a_1,a_2$). To illustrate this, consider the range of $\alpha_{ext}$ values (0.29 - 3.25) found from linear fits over 0.3 - 0.7 μm to all of the spectra measured by SpEx during KORUS-OC. This range of $\alpha_{ext}$ values maps differently into ($a_1,a_2$) space as a function of $\lambda_{ch}$ (Fig. 9, top right panel).

There are two special cases evident in Eq. (5) that result in $\alpha_{ext}$ = $a_1$. First, when there is no curvature ($a_2$ = 0), $a_1$ describes the same linear fit as $\alpha_{ext}$. Second, when $\lambda_{ch}$ = 1 μm (i.e., LN(1) = 0) Eq. (5) is insensitive to curvature such that $a_2$ can have any value at all. This can be understood from Eq. (1), where $\alpha$ can be any value when $\lambda$ = 1 μm and p(1μm) will always = β. The former leads to all $\lambda_{ch}$ sets overlapping at $a_2$ = 0, while the latter exhibits a broad vertical band independent of curvature ($a_2$) (Fig. 9, top

right panel). These special cases have important implications. As $\lambda_{ch}$ approaches 1 µm the measurement becomes insensitive to curvature, while at the short wavelengths of light represented by $\lambda_{ch} < 0.1$ µm the curvature itself becomes unimportant. Hence, to probe spectral curvature the upper right panel of Fig. 9 shows measurement techniques with $\lambda_{ch} \sim 0.5 \pm 0.2$ µm provide the greatest sensitivity with sufficient separation in $(a_1, a_2)$ to distinguish aerosol microphysical and chemical properties influencing the spectral shape.

The rotation as a function of $\lambda_{ch}$ shown in Fig. 9 also illustrates why wavelength units of µm must be used to calculate $(a_1, a_2)$. As $\lambda_{ch}$ increases to values > 1 µm, the $\alpha$ map rotates clockwise (Fig. S8). If one used $\lambda_{ch} = 410$ nm rather than 0.41 µm, it would map into a narrow band in the next quadrant of $(a_1, a_2)$ space spanning a wide range in $a_1$ but a narrow range in $a_2$, resulting in little curvature sensitivity. The calculation of $\alpha_{ext}$ is wavelength independent and will produce the same result no matter what units are used. This is *not* the case for the calculation of $(a_1, a_2)$, so it must be emphasized that for curvature, the units matter.

The angular difference between $\lambda_{ch} = 0.61$ µm and 0.41 µm accounts for the shift between the mapping reported in Schuster et al. (2006) and this work. The differing values in $\lambda_{ch}$ arise from the different spectral ranges between AERONET (7 bands spanning 0.34 - 1.02 µm, Schuster et al. (2006)) versus 0.3 - 0.7 µm for SpEx. The $\alpha_{ext}$ values from fits to the spectra set map into the expected bands (Fig. 9, bottom panel) with the color distribution shifting slightly over the range of calculated $\lambda_{ch} = 0.36 - 0.46$ µm. Note, that the most extreme values in $a_1$ for the KORUS-OC data set are related to shorter $\lambda_{ch}$ than the rest of the data set.

Schuster et al. (2006) used this type of coefficient mapping to distinguish different aerosol size distributions via curvature that otherwise exhibit the same $\alpha$. In particular, while fine mode aerosols exhibit negative curvature, the presence of sufficient coarse mode aerosols in a bimodal size distribution induces positive curvature due to the efficient extinction of light at longer wavelengths by larger particles. Here, the inlet limited the size range of sampled aerosol to the submicron fraction, such that positive curvature is not expected. Aerosol size distributions were not measured aboard ship during the cruise but as described in Sect. 3.1, previously published work provides sufficient information for a broad characterization of the different ambient aerosol populations prevalent during the three meteorological regimes that occurred during KORUS-OC. This context is used to assess the mapping of SpEx data into $(a_1, a_2)$ space. For clarity, the 60 min mean spectra data are used (Fig. 10).

In addition to the evident separation in $\alpha_{ext}$ across $(a_1, a_2)$ space (Fig. 9 and top left panel of Fig. 10) there is also clear separation as a function of aerosol loading using $\sigma_{ext}(0.532$ µm) as a proxy for ambient aerosol concentrations (Fig. 10, lower left panel). High concentrations ($\sigma_{ext}(0.532$ µm) > 150 Mm$^{-1}$) exhibit a relatively small range of $a_1$ and $a_2$ values. Generally, these spectra exhibit the greatest curvature (i.e., largest absolute values of $a_2$ for any given $a_1$). In contrast, low concentrations ($\sigma_{ext}(0.532$ µm) < 75 Mm$^{-1}$) span a wide range of values in $a_1$ and $a_2$. As described in Sect. 3.1 there were three distinct meteorological regimes during the cruise (Peterson et al., 2019) that led to different ambient aerosol populations. Hence, the separation in aerosol loading should not be viewed as a function of loading for a uniform aerosol population, but rather as an artifact of the differing size distributions and to a lesser extent, composition.

The $(a_1, a_2)$ map separated according to the defined meteorological regimes reveals strikingly different distributions for the 3 periods (Fig. 10, top right panel). The spectra during the Stagnant period (predominantly submicron aerosols, where PM$_{2.5}$ ~ PM$_1$ (Jordan et al., 2020a), dominated by locally produced SOA (Kim et al., 2018; Nault et al., 2018; Peterson et al., 2019; Choi et al., 2019;

Jordan et al., 2020a)) produced a remarkably narrow range of $a_1$ and $a_2$ values that essentially lie along a single $\alpha_{ext}$ line (~1.5) for the 60 min mean spectra set. In contrast, the spectra during the Blocking period (likely small absorbing aerosols from relatively fresh ship emissions) exhibit a wide range in $a_1$ and $a_2$ values with values of $\alpha_{ext}$ generally > 1.6. However, $\alpha_{ext}$ values for this group also span the full range of observed $\alpha_{ext}$ primarily when the absolute value of $a_2$ was small ($\leq |-0.6|$). The large variability of this group may be due in part to the low extinctions where the sensitivity of $\alpha_{ext}$ to uncertainty in $\sigma_{ext}$ is greatest. It may also reflect the heterogeneity of aerosol sources encountered in the marine boundary layer as the ship cruised around the West Sea during this period.

Finally, the period when aerosol concentrations were highest, Transport/Haze, exhibits the same range in $a_1$ as the Stagnant aerosols, but with greater curvature (i.e., larger absolute values of $a_2$, Fig. 10, top right panel). During this period the meteorological conditions that transported polluted air masses eastward from China, also created conditions that promoted rapid secondary inorganic aerosol production locally over the S. Korean peninsula (Peterson et al., 2019; Eck et al., 2020; Jordan et al., 2020a) that resulted in the growth of fine mode aerosols to larger sizes ($PM_{2.5} > PM_1$, Eck et al., 2020; Jordan et al., 2020a) than observed during the Stagnant period. During the first half of the Transport/Haze period the ship was downwind of S. Korea in the East Sea, while during the second half it was upwind in the West Sea. Splitting the data from this period to reflect the position of the ship (Fig. 10 bottom right panel) shows that the highest concentrations of aerosols were observed downwind of S. Korea and exhibited the greatest curvature and the lowest $\alpha_{ext}$. This result indicates larger particle sizes were present downwind than upwind, consistent with the reported changes in aerosol size distribution due to local production over the Korean peninsula (Eck et al., 2020; Jordan et al., 2020a). The upwind distribution resembles the narrow Stagnant distribution in $(a_1,a_2)$ space, but shifted to a slightly lower $\alpha_{ext}$.

It is interesting to contrast the range in curvature between the three periods. As shown in Fig. 6, $\alpha_{ext}$ decreases as the fit range is extended to shorter wavelengths. This is due to the curvature evident in the UV range of Fig. 7 which is not adequately captured by fits to the longer wavelength subranges. In that instance (an individual spectrum from the Stagant period), $\alpha_{ext}(0.45\text{-}0.632\ \mu m) = 3$, whereas $\alpha_{ext}(0.3\text{-}0.7\ \mu m) = 2.04$. The greatest curvature tends to be found for the period with the largest particles observed during the campaign, while for the period when the particles were likely to be smallest, the UV curvature is small or absent leading to larger $\alpha_{ext}$ values (Fig. 10). In addition, the largest absolute values of $a_1$ ($\geq |-6|$) found in the individual spectra (Fig. 9) arise from partial spectra where the longer wavelengths of a measured spectrum are below detection. Spectral fits were limited to only above detection portions of the measured spectrum. This is why the $\lambda_{ch}$ for these spectra shift to shorter wavelengths. These spectra are those for which scattering and extinction were observed to be low, hence, the spectral fit is subject to greater uncertainty. This accounts for the finding that some of the fine fraction Blocking period aerosols, both exhibit curvature as large as the other two periods, as well as the limited negative curvature ($a_2 \leq |-0.5|$) and in a few cases, slightly positive curvature ($a_2 \geq 0$, Figs. 9 and 10). Note, partial spectra are not suitable for retrievals (i.e., comparable to those from AERONET Level 2 data where at a minimum above detection values must be available from at least the 0.38, 0.50, and 0.87 $\mu m$ channels to ensure nonlinearity in the spectrum is adequately represented). However, partial spectra can be valuable for other analyses such as when combined with absorption coefficients in the calculation of $\omega(\lambda)$ to look for structure in the above detection range for SpEx, particularly in the UV (see Part 2, Jordan et al., 2020b). Hence, partial spectra data are not discarded from further examination.

**5 Conclusions**

This work, Part 1 of 2, examined the high temporal resolution data set (IN101, TAP, and SpEx) collected as part of the in situ aerosol measurement suite deployed aboard the *R/V Onnuri* for the KORUS-OC cruise. IN101 scattering ($\sigma_{scat}$) and TAP absorption ($\sigma_{abs}$) coefficients were measured at three visible wavelengths throughout the cruise, with single scattering albedo ($\omega$) calculated from them. These data were presented to provide an overview of the in situ aerosol measurements throughout the cruise within the context of the prevalent meteorological regimes previously reported for the KORUS-AQ field campaign (Peterson et al., 2019; Jordan et al., 2020a). The cruise took place during 3 distinct meteorological periods where, 1) stagnant conditions fostered local (S. Korean) production of secondary organic aerosol, 2) transport from China coupled with local overcast and humid hazy conditions led to secondary production of inorganic aerosol with rapid growth of fine mode aerosols to larger particle sizes, and 3) a blocking period with limited transport following a frontal passage that dramatically reduced aerosol concentrations. Results presented here suggest the aerosols observed aboard *R/V Onnuri* during this final period were likely relatively fresh small particles from ship emissions into the marine boundary layer. The largest values of $\sigma_{scat}$ and $\sigma_{abs}$ were observed when the ship was downwind of the Korean peninsula in the East Sea during the Transport/Haze period. The smallest values of $\omega$ were found when the ship was upwind of the peninsula in the West Sea, with low values arising from reductions in scattering rather than increases in absorption.

Extinction coefficients ($\sigma_{ext}$) calculated from the 3 visible wavelength $\sigma_{scat}$ and $\sigma_{abs}$ data were used to evaluate the performance of SpEx under field conditions that offered a wide range of concentrations, particles sizes, and composition. Excellent agreement was found for all 3 wavelengths with slopes equal to $1.020 \pm 0.002$, $0.998 \pm 0.003$, and $1.057 \pm 0.004$ with $r^2 = 0.981$, 0.969, and 0.942 for the 450, 532, and 632 nm channels, respectively. A lower limit of detection of 10 Mm$^{-1}$ was determined for the individual spectral measurements (twice the standard deviation of the measurement) that can be reduced via standard error of the means when averaging spectra over longer sampling intervals. The broad spectral range (300 - 700 nm) and fine spectral resolution (0.8 nm) provided an opportunity to examine the wavelength dependence of the spectra for a diverse set of in situ ambient aerosols. Nearly all of the measured spectra exhibited curvature in logarithmic space such that 2$^{nd}$ order polynomials provided a better fit to the data than the usual linear fit of a power law representation. With either fit, evidence was presented to highlight the large deviation of an extrapolated value for $\sigma_{ext}$ beyond the fit wavelength range. This finding highlights the need for measurements that extend well into the UV, thereby limiting the need for extrapolated estimates of $\sigma_{ext}$ in that part of the spectrum.

A comparison to a previous study of spectral curvature based on Mie calculations and remote sensing data from AERONET (Schuster et al., 2006) revealed the wavelength dependence that relates the Angstrom exponent ($\alpha$) to the 2$^{nd}$ order polynomial coefficients ($a_1$ and $a_2$). The characteristic wavelength ($\lambda_{ch}$) of any given data set needs to be taken into account when comparing spectral curvature coefficients across data sets. Mapping the fit coefficients shows that any given $\alpha$ representation can be separated along a line in ($a_1, a_2$) with a slope of $-2LN(\lambda_{ch})$ such that spectral curvature can be used to obtain more detailed information about aerosol size distribution. The work of Schuster et al. (2006) was directed to distinguishing different bimodal size distributions on the basis of the presence of coarse fraction aerosols. Here, only fine mode aerosols were sampled, nonetheless, the separation found in ($a_1, a_2$) space across the KORUS-OC data set suggests that curvature may be used to infer more detailed size distribution information even within the fine mode alone. Size distributions were not measured aboard the *R/V Onnuri*, so such a study will require future ambient measurements to fill this data gap.

In Part 2 (Jordan et al., 2020b), the methodology used for the filter analyses from the KORUS-OC in situ aerosol measurement suite is described with an overview of the results provided. The data from those filters include total aerosol $\sigma_{abs}$ spectra (300-700 nm) from glass fiber filters placed in the center of an integrating sphere, soluble aerosol absorption coefficient spectra (300-700 nm) from deionized water ($\sigma_{DI-abs}$) and methanol ($\sigma_{MeOH-abs}$) extracts of Teflon filters measured with a liquid waveguide capillary cell, water-soluble inorganic ion (WSII) concentrations via ion chromatography, and water-soluble organic compounds (WSOC) that contribute to the aerosol measured using an aerosol mass spectrometer. The combination of filter-based $\sigma_{abs}$ spectra (300-700 nm) with the SpEx $\sigma_{ext}$ spectra set, allows for the calculation of spectral $\omega$ (300-700 nm) for in situ aerosols. Part 2 includes a similar examination of power law and $2^{nd}$ order polynomial representations of all 4 of the in situ aerosol hyperspectral data sets obtained during KORUS-OC. It also explores relationships between the optical properties and water-soluble composition information within the meteorological context of KORUS-AQ following the discussion presented here in Part 1.

**Data Availability**

All data presented here are available under the *R/V Onnuri* Ship tab in the KORUS-AQ archive (DOI: 10.5067/Suborbital/KORUSAQ/DATA01).

**Author contribution**

CEJ led the experiment, analyzed the data, and wrote the manuscript.
CEJ, BEA, LDZ, CHH, KLT, ELW, RFM, MAS, AJB, CER built elements of the hardware, software, and deployment measurement system; and assisted in the laboratory at NASA LaRC.
CEJ, BEA, AJB, CAC participated in the field work.
RMS, BTL, & MAT deployed with the measurement suite aboard the *R/V Onnuri*, collected filter samples, and contributed to the manuscript.
GLS, RHM, LDZ, BEA, ECC, MAS, RMS, AJB, & CAC contributed to the data analysis and the manuscript.

**Competing interests**

The authors declare that they have no conflict of interest.

**Acknowledgements**

The authors gratefully acknowledge the support of the KORUS-OC and KORUS-AQ science teams, the outstanding support provided by our South Korean partners at the Korean Institute for Ocean Science and Technology (KIOST), and financial support from the NASA/NIA cooperative agreement NNL09AA00A and NASA Grant NNX16AD60G through the Geostationary Coastal and Air Pollution Events (GEO-CAPE) mission pre-formulation studies. The authors particularly thank Anne Thompson for her support throughout this study and Fred Brechtel and Vanderlei Martins for helpful discussions

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

**Figure 1:** Cruise maps of date and time (KST, top left) with 532 nm $\sigma_{scat}$ (Mm$^{-1}$, top right), $\sigma_{abs}$ (Mm$^{-1}$, bottom left) and $\omega$ (bottom right).

**Figure 2:** Top: $\sigma_{scat}$ (light shades) and $\sigma_{abs}$ (dark shades) averaged to 1 minute intervals during the cruise (both in units of Mm$^{-1}$). Middle: $\alpha_{scat}$ (light shades) and $\alpha_{abs}$ (dark shades), unitless, calculated from the wavelength pairs of the coefficients in the top panel. Bottom: $\omega$ (right axis), unitless, along with 1 minute averaged CO concentrations (ppbv). Note, the peak CO value cut off in the figure reached 1000 ppbv.

**Figure 3:** NT vs. SpEx $\sigma_{ext}$ (450 nm, top; 532 nm, middle; 632 nm bottom) All data points (gray pluses) are shown with 15 min (light colored circles), 30 min (dark colored triangles), and 60 min (black symbols) means. Fit lines, coefficients, and $r^2$ values are color-coordinated with the symbols, except for the black markers where the fit lines used a light color for visibility. Above LLOD points only.

**Figure 4:** Time series of 532 nm $\sigma_{ext}$ (Mm$^{-1}$) throughout the cruise. Top panel: SpEx (all data, gray; above LLOD, green; these curves are coincident until June 2$^{nd}$ when the lowest values are below detection and hence, appear gray) with NT $\sigma_{ext}$ (black). Middle panel: SpEx (all data, gray) with 15 min (light green), 30 min (dark green), and 60 min (black) means. Bottom panel: SpEx standard deviations of the 15 min (light green), 30 min (dark green), and 60 min (black) means. Meteorological periods shown as in Fig. 2.

**Figure 5:** Two day highlight of the top two panels of Fig. 4. Top panel: $\sigma_{ext}$(532 nm) from SpEx (green) shown with NT (black). Bottom panel: $\sigma_{ext}$(532 nm) from SpEx (all data, gray) with 15 min (light green), 30 min (dark green), and 60 min (black) means.

**Figure 6:** $\alpha_{ext}$ determined over 3 different wavelength ranges (300-700 nm, black diamonds, 450-532 nm, teal circles, and 532-632 nm, gold triangles) compared to that found over the 450-632 nm range (x-axis). All data (left panel) and 30 min means (right).

**Figure 7.** Example of wavelength dependence of LN($\sigma_{ext}$ (Mm$^{-1}$)) spectra (red curves) as a function of LN(wavelength (μm)). Linear (y = a + b(x), left panels; here the intercept a = LN($\beta$), the value of LN($\sigma_{ext}$) at 1 μm where LN(1 μm) = 0, and the slope b = -$\alpha$) and 2$^{nd}$ order polynomial (y = $a_0$ + $a_1$(x) + $a_2$($x^2$), right panels) fits (black curves) are shown with the fit residuals (= LN($\sigma_{ext}$ (Mm$^{-1}$)) - fit, blue curves). Residuals randomly distributed around zero indicate a good fit by the mathematical function used to fit the data, trends in residuals suggest another function may provide a better fit. Top and bottom panels show fits to a subrange (LN(0.450 - 0.632 μm)) and full range (LN(0.3 - 0.7 μm) of the measured spectrum, respectively. Middle panels show the extrapolation of the fit in the top panels over the full measured wavelength range. The x-axis labels of -1.2, -1.0, -0.8, -0.6 and -0.4 for LN($\lambda$ (μm)) equal 0.301, 0.368, 0.449, 0.549, 0.670, and 0.698 um wavelengths, respectively.

**Figure 8:** Comparison of residuals (the difference between the data and mathematical function fit to that data) from a line fit (black) and 2$^{nd}$ order polynomial fit (red) to the measured LN($\sigma_{ext}$(Mm$^{-1}$)) spectra over the 0.3 - 0.7 μm range for 6 wavelengths: 0.315 (top left), 0.365 (top right), 0.45 (middle left), 0.532 (middle right), 0.632 (bottom left), and 0.675 μm (bottom right).

**Figure 9:** Top left: coefficients $a_2$ versus $a_1$ from 2$^{nd}$ order polynomial fits to the full wavelength range (0.3 - 0.7 μm) of the individual (All Data, gray filled circles) and mean (15 min, red open circles, 30 min, dark blue triangles, and 60 min, light blue diamonds, average) spectra. The black box shows the limits of the Schuster et al. [2006] Fig. 6 plot, black lines show approximate equivalents to $\alpha_{ext}$ = 1 and 2 from that work. Top right: $\alpha$ mapped into ($a_1$,$a_2$) space as a function of the characteristic wavelength ($\lambda_{ch}$) of the measured spectral range. Bottom: $\alpha_{ext}$ calculated from the full spectral range of SpEx (colored dots) overlaid on $\alpha$ maps that cover the range of $\lambda_{ch}$ values calculated from the data set. The black box is the same as the one in the top left panel.

**Figure 10:** Coefficients $a_2$ vs. $a_1$ from 2$^{nd}$ order polynomial fits to the full wavelength range (0.3-0.7 μm) of the 60 min mean spectra colored by $\alpha_{ext}$ ($\lambda$=0.3-0.7 μm), top left, $\sigma_{ext}$ (0.532 μm (Mm$^{-1}$)), bottom left, and the defined meteorological periods described in Peterson et al. [2019], top right (i.e., excluding the interval when the meteorological regime was in transition, May 23$^{rd}$ and 24$^{th}$, and hence, undefined). The bottom right panel is the same as the top right, but with the Transport/Haze samples split between those measured to the east (May 25$^{th}$-27$^{th}$) and west (May 29$^{th}$-31$^{st}$) of the peninsula, excluding those samples collected in transit between the two.

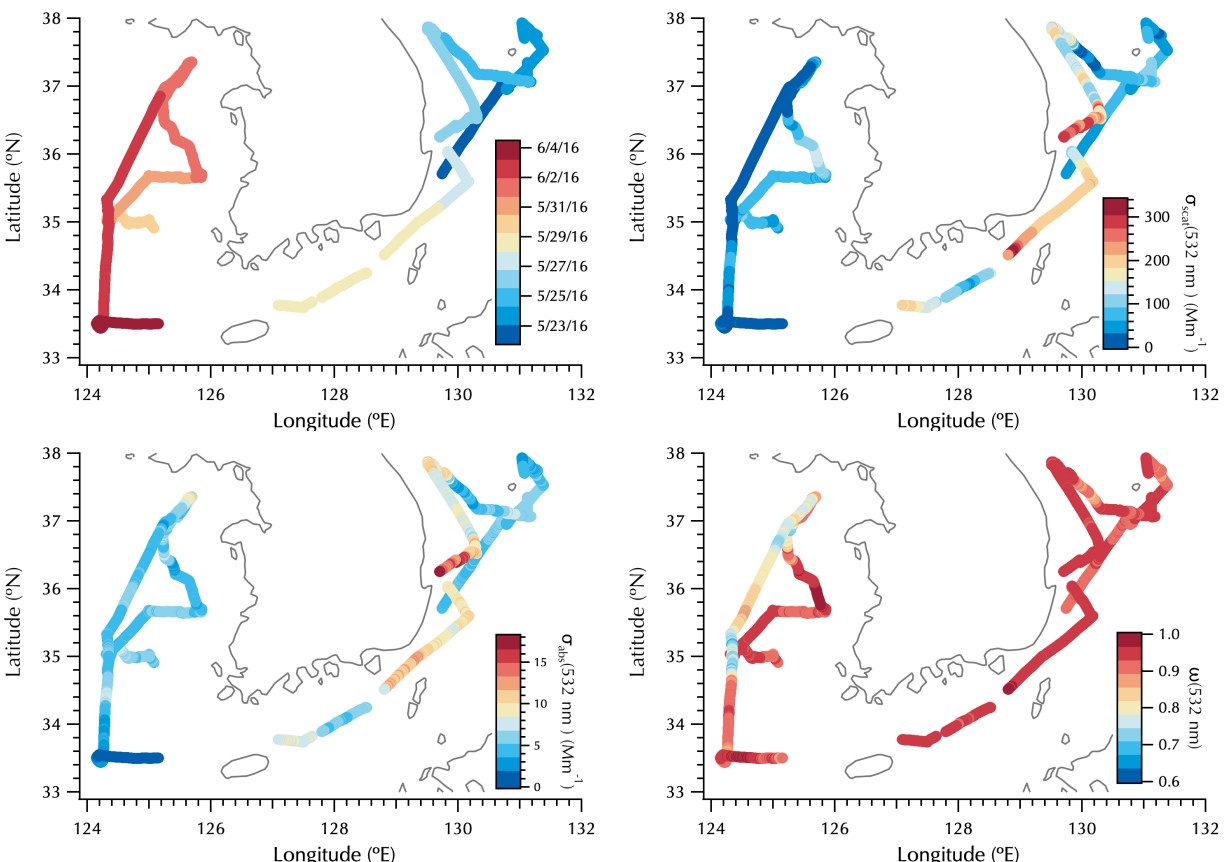

**Figure 1: Cruise maps of date and time (KST, top left) with 532 nm $\sigma_{scat}$ (Mm$^{-1}$, top right), $\sigma_{abs}$ (Mm$^{-1}$, bottom left) and $\omega$ (bottom right).**

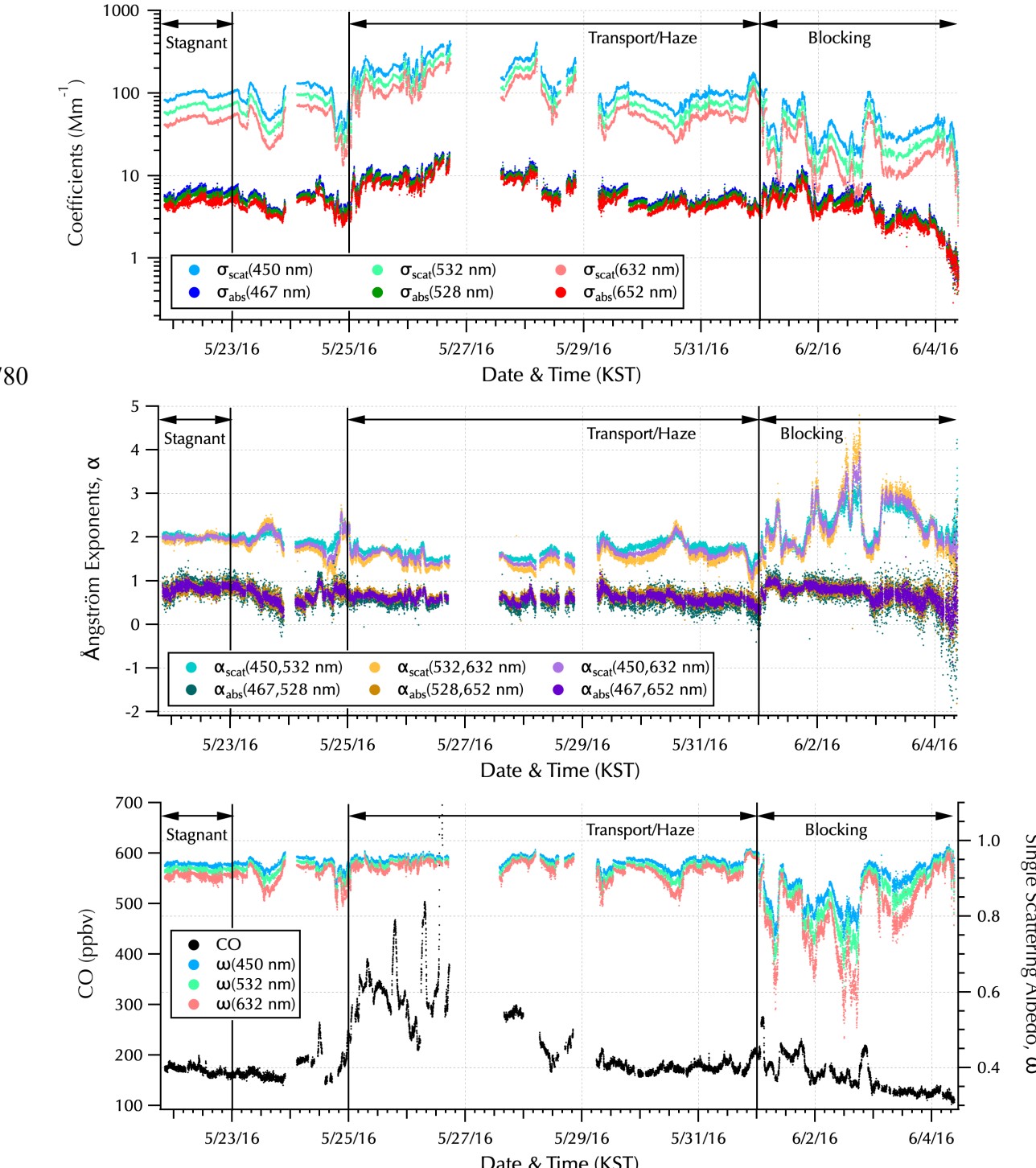

**780**

**Figure 2:** Top: $\sigma_{scat}$ (light shades) and $\sigma_{abs}$ (dark shades) averaged to 1 minute intervals during the cruise (both in units of Mm$^{-1}$). Middle: $\alpha_{scat}$ (light shades) and $\alpha_{abs}$ (dark shades), unitless, calculated from the wavelength pairs of the coefficients in the top panel. **785** Bottom: $\omega$ (right axis), unitless, along with 1 minute averaged CO concentrations (ppbv). Note, the peak CO value cut off in the figure reached 1000 ppbv.

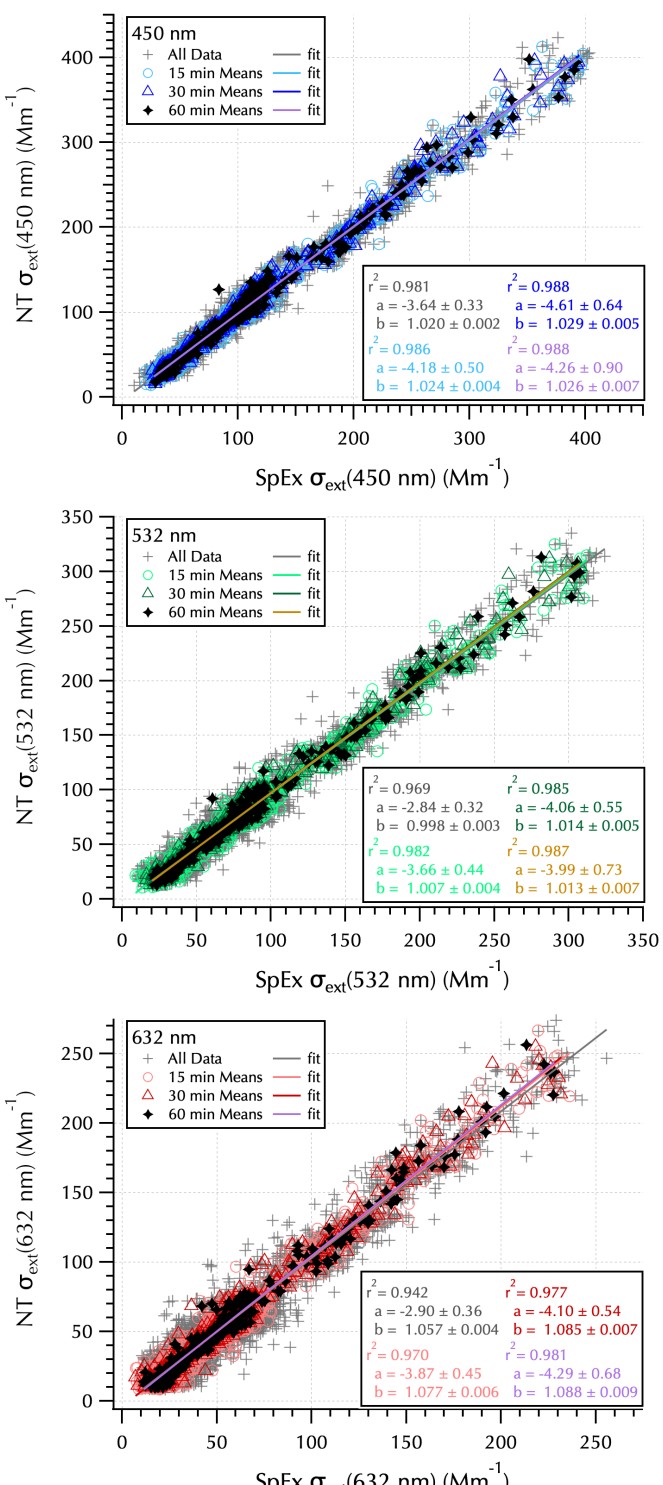

**Figure 3: NT vs. SpEx $\sigma_{ext}$ (450 nm, top; 532 nm, middle; 632 nm bottom)** All data points (gray pluses) are shown with 15 min (light colored circles), 30 min (dark colored triangles), and 60 min (black symbols) means. Fit lines, coefficients, and $r^2$ values are color-coordinated with the symbols, except for the black markers where the fit lines used a light color for visibility. Above LLOD points only.

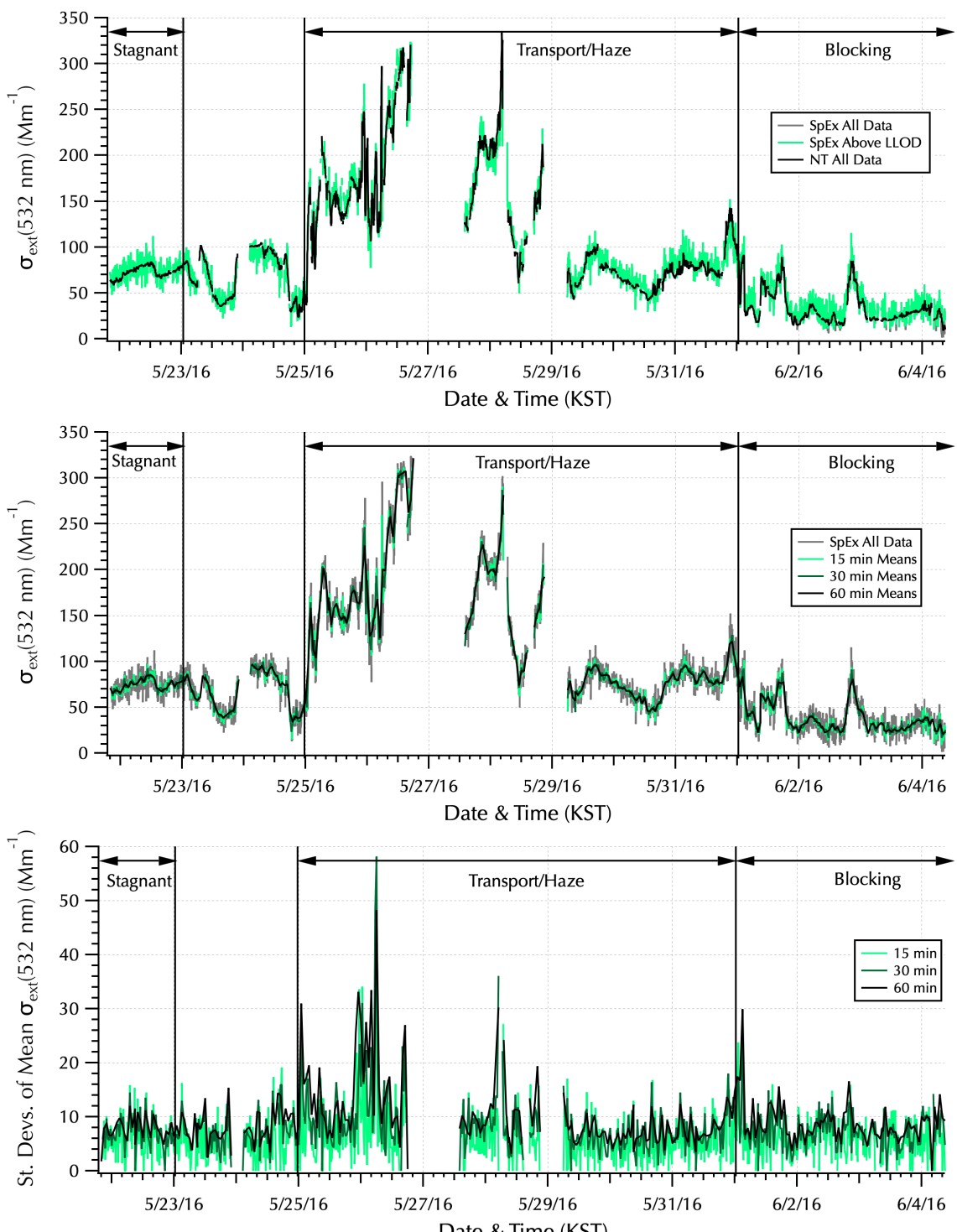

**Figure 4: Time series of 532 nm σ$_{ext}$ (Mm$^{-1}$) throughout the cruise. Top panel: SpEx (all data, gray; above LLOD, green; these curves are coincident until June 2$^{nd}$ when the lowest values are below detection and hence, appear gray) with NT σ$_{ext}$ (black). Middle panel: SpEx (all data, gray) with 15 min (light green), 30 min (dark green), and 60 min (black) means. Bottom panel: SpEx standard deviations of the 15 min (light green), 30 min (dark green), and 60 min (black) means. Meteorological periods shown as in Fig. 2.**

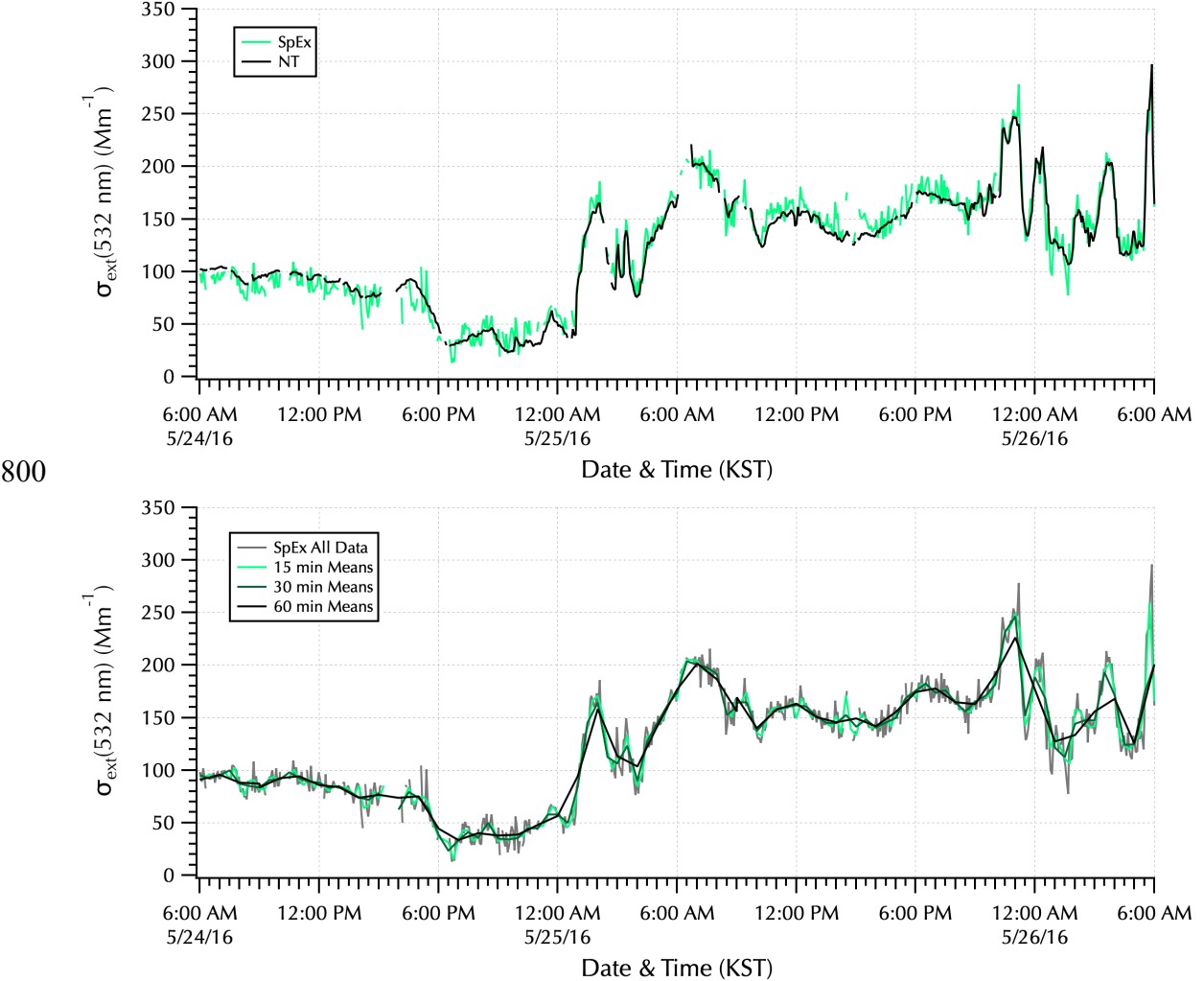

**Figure 5: Two day highlight of the top two panels of Fig. 4. Top panel: $\sigma_{ext}$(532 nm) from SpEx (green) shown with NT (black). Bottom panel: $\sigma_{ext}$(532 nm) from SpEx (all data, gray) with 15 min (light green), 30 min (dark green), and 60 min (black) means.**

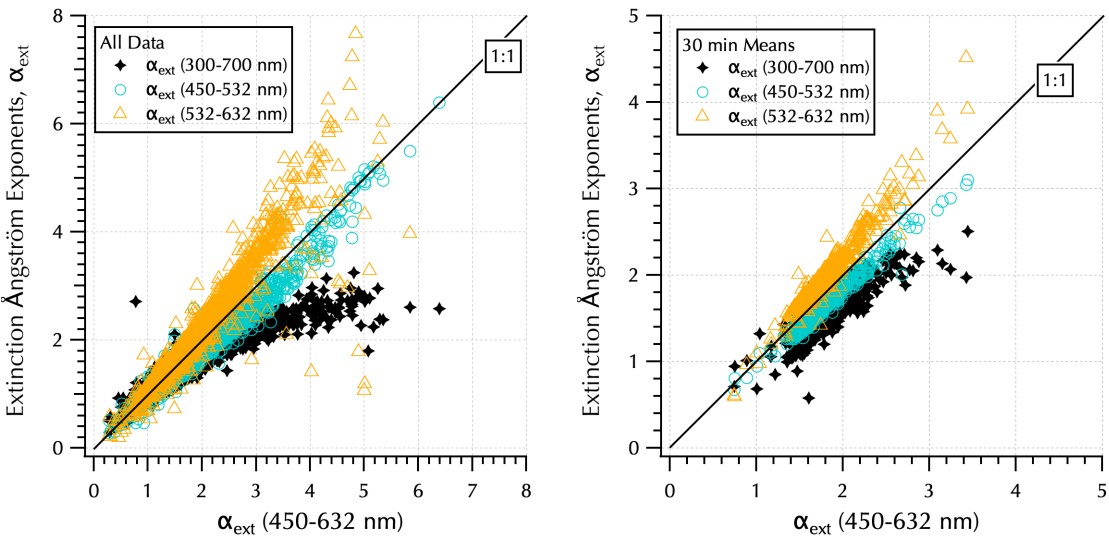

**Figure 6:** $\alpha_{ext}$ determined over 3 different wavelength ranges (300-700 nm, black diamonds, 450-532 nm, teal circles, and 532-632 nm, gold triangles) compared to that found over the 450-632 nm range (x-axis). All data (left panel) and 30 min means (right).

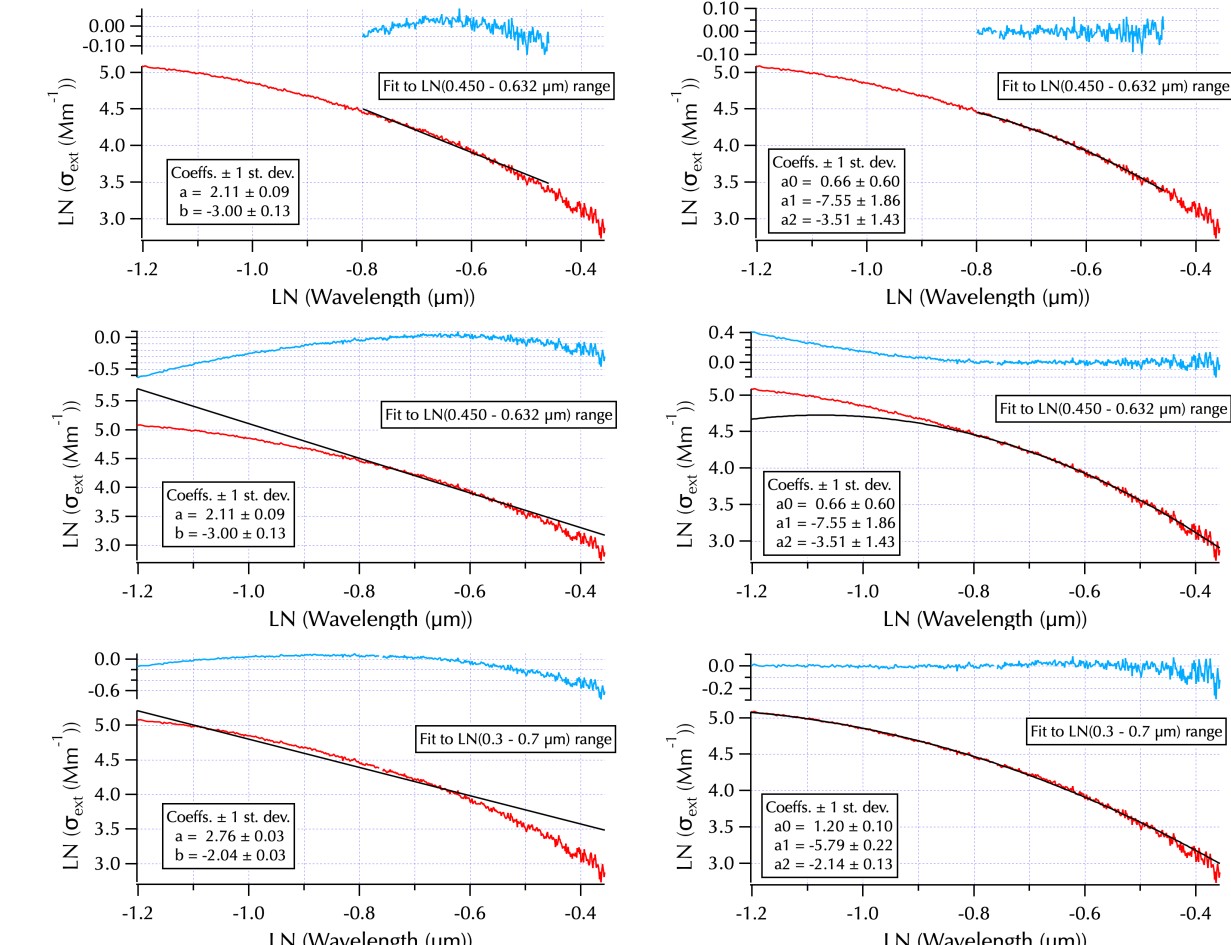

**Figure 7. Example of wavelength dependence of LN($\sigma_{ext}$ (Mm$^{-1}$)) spectra (red curves) as a function of LN(wavelength (μm)). Linear (y = a + b(x), left panels; here the intercept a = LN(β), the value of LN($\sigma_{ext}$) at 1 μm where LN(1 μm) = 0, and the slope b = -α) and 2$^{nd}$ order polynomial (y = a$_0$ + a$_1$(x) + a$_2$(x$^2$), right panels) fits (black curves) are shown with the fit residuals (= LN($\sigma_{ext}$ (Mm$^{-1}$)) - fit, blue curves). Residuals randomly distributed around zero indicate a good fit by the mathematical function used to fit the data, trends in residuals suggest another function may provide a better fit. Top and bottom panels show fits to a subrange (LN(0.450 - 0.632 μm)) and full range (LN(0.3 - 0.7 μm) of the measured spectrum, respectively. Middle panels show the extrapolation of the fit in the top panels over the full measured wavelength range. The x-axis labels of -1.2, -1.0, -0.8, -0.6 and -0.4 for LN(λ (μm)) equal 0.301, 0.368, 0.449, 0.549, 0.670, and 0.698 um wavelengths, respectively.**

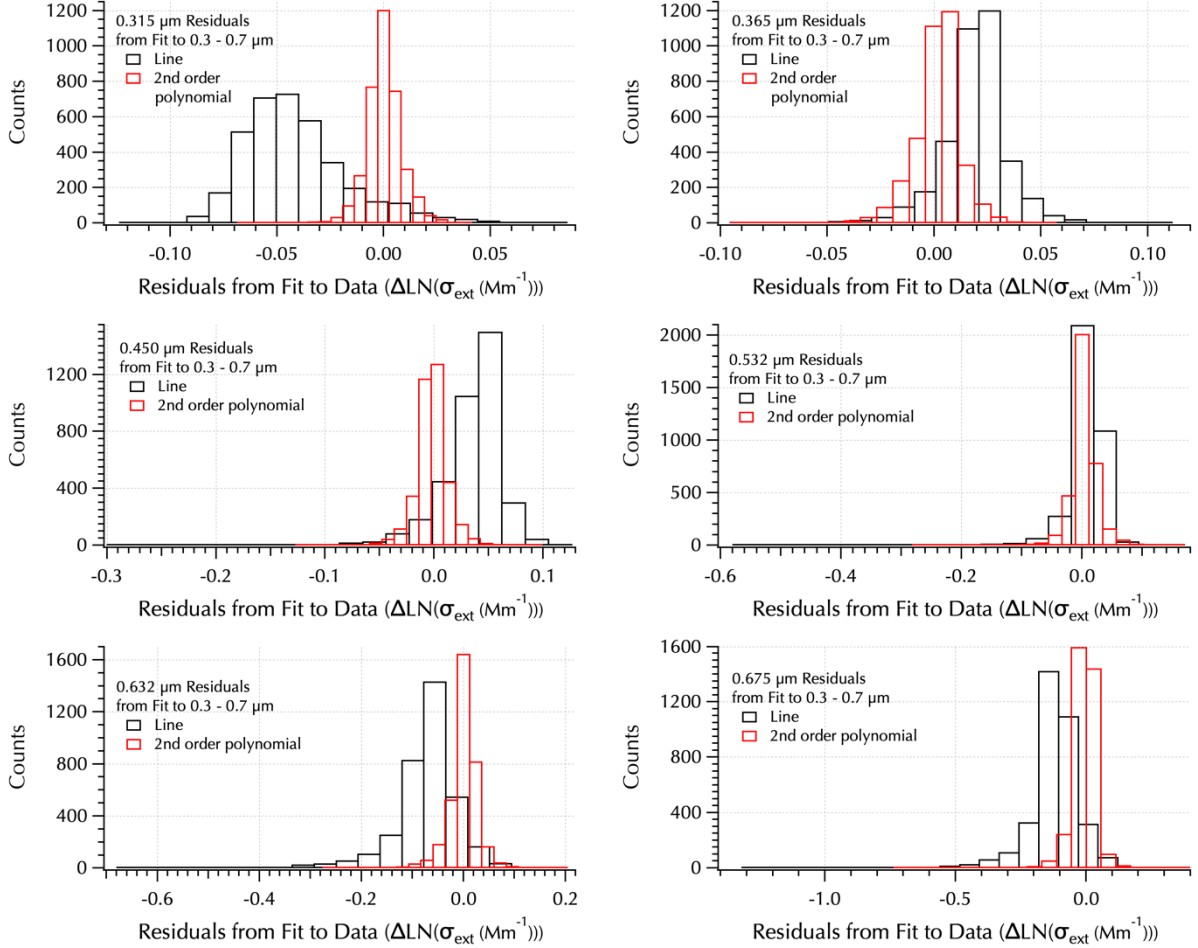

**Figure 8:  Comparison of residuals (the difference between the data and mathematical function fit to that data) from a line fit (black) and 2nd order polynomial fit (red) to the measured LN(σ_ext(Mm⁻¹)) spectra over the 0.3 - 0.7 μm range for 6 wavelengths: 0.315 (top left), 0.365 (top right), 0.45 (middle left), 0.532 (middle right), 0.632 (bottom left), and 0.675 μm (bottom right).**

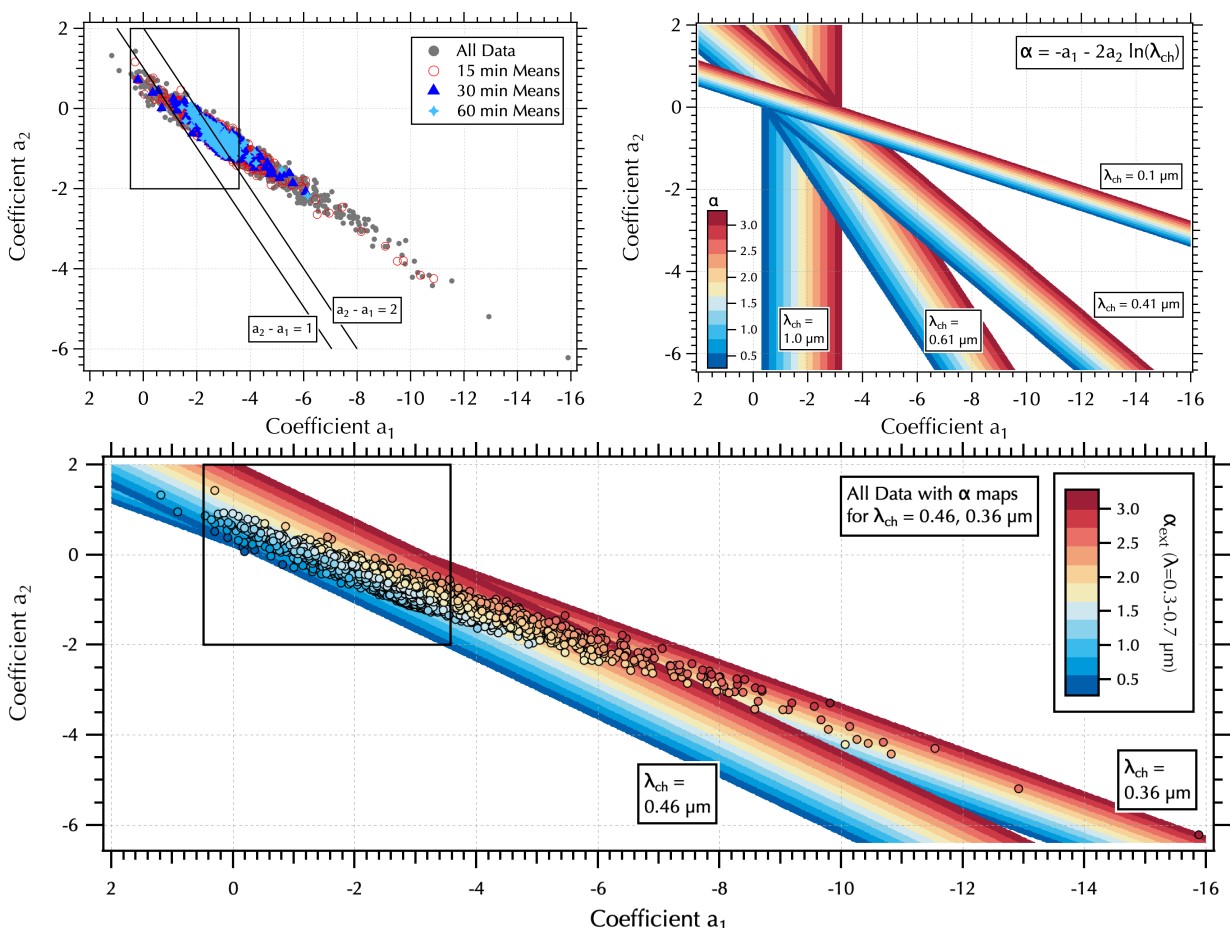

**Figure 9:** Top left: coefficients $a_2$ versus $a_1$ from 2$^{nd}$ order polynomial fits to the full wavelength range (0.3 - 0.7 μm) of the individual (All Data, gray filled circles) and mean (15 min, red open circles, 30 min, dark blue triangles, and 60 min, light blue diamonds, average) spectra. The black box shows the limits of the Schuster et al. [2006] Fig. 6 plot, black lines show approximate equivalents to $\alpha_{ext}$ = 1 and 2 from that work. Top right: $\alpha$ mapped into ($a_1$,$a_2$) space as a function of the characteristic wavelength ($\lambda_{ch}$) of the measured spectral range. Bottom: $\alpha_{ext}$ calculated from the full spectral range of SpEx (colored dots) overlaid on $\alpha$ maps that cover the range of $\lambda_{ch}$ values calculated from the data set. The black box is the same as the one in the top left panel.

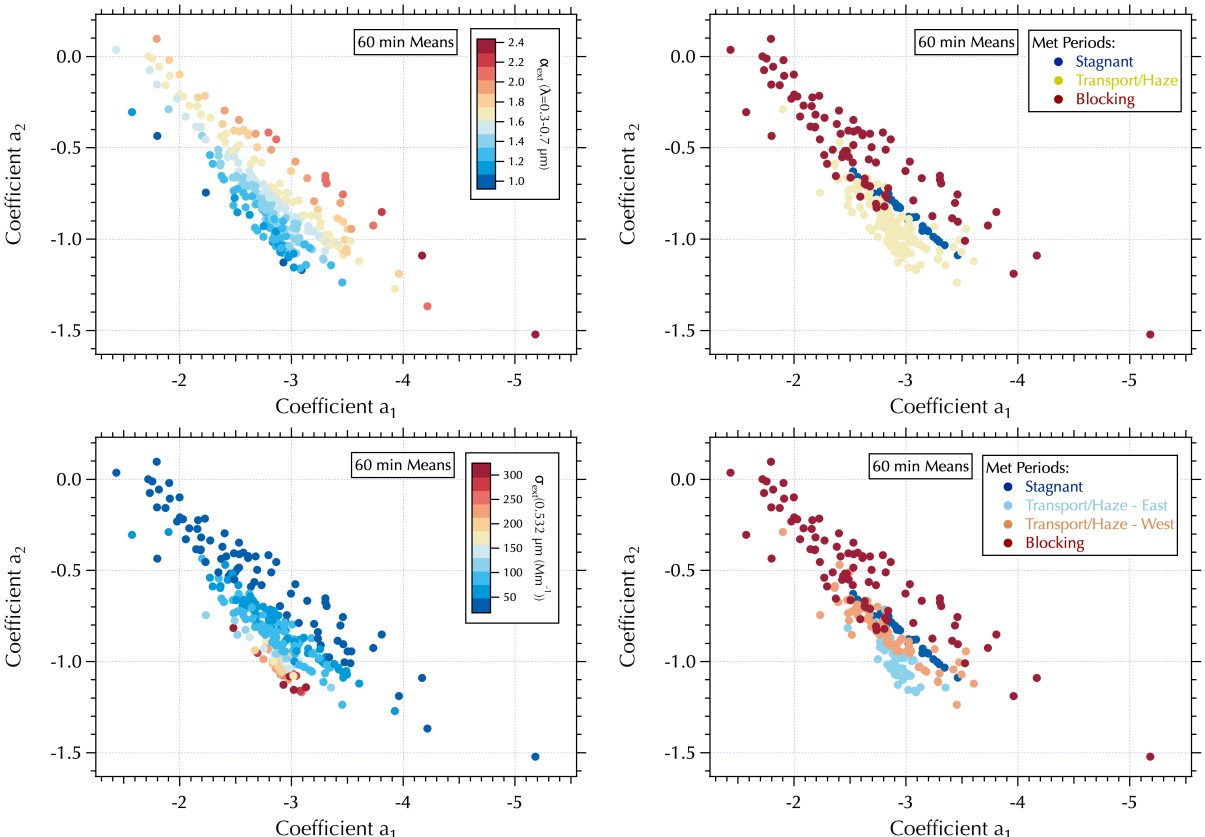

**Figure 10:** Coefficients $a_2$ vs. $a_1$ from $2^{nd}$ order polynomial fits to the full wavelength range (0.3-0.7 μm) of the 60 min mean spectra colored by $\alpha_{ext}$ (λ=0.3-0.7 μm), top left, $\sigma_{ext}$ (0.532 μm (Mm$^{-1}$)), bottom left, and the defined meteorological periods described in Peterson et al. [2019], top right (i.e., excluding the interval when the meteorological regime was in transition, May $23^{rd}$ and $24^{th}$, and hence, undefined). The bottom right panel is the same as the top right, but with the Transport/Haze samples split between those measured to the east (May $25^{th}$-$27^{th}$) and west (May $29^{th}$-$31^{st}$) of the peninsula, excluding those samples collected in transit between the two.