# Peer review of "New In Situ Aerosol Hyperspectral Optical Measurements over 300-700 nm, Part 1: Spectral Aerosol Extinction (SpEx) Instrument Field Validation during the KORUS-OC cruise"

_Atmospheric Measurement Techniques, 2020_

## Referee Comment (RC1) · Anonymous Referee #1 · 2 Sep 2020

This paper describes the first field measurements by the spectral aerosol extinction (SpEx) instrument. The SpEx instrument measures broadband aerosol extinction at 300 - 700 nm with 0.8 nm resolution. It was deployed on a ship cruise around the Korean peninsula during May – June 2016.

The paper presents Angstrom exponents and second-order polynomial fits to the extinction data, and determines that second-order polynomial fits are a better representation.

[Figure]

This paper builds on previous instrument development work described in Jordan et al 2015 AMT and Chartier and Greenslade 2012 AMT.

Major comments:

1. The manuscript would be improved by examining the underlying relationship between aerosol extinction and wavelength.

As described in the introduction, the major intrinsic properties that determine aerosol extinction are the aerosol size distribution, aerosol shape, and complex refractive index. The Angstrom exponent, A (where Extinction = k wavelengthˆ-A) is an empirical expression to describe the wavelength-dependence of the aerosol extinction. Some combinations of size distribution, shape, and complex refractive index produce extinction values that are not well-represented by an Angstrom exponent.

This paper presents Angstrom exponents and an alternative second-order polynomial fit to the extinction data. The paper would be much stronger if the authors calculated the expected aerosol extinction from the size distribution, assumed shape, and complex refractive index (volume-weighted by composition). They could then describe the types of aerosol populations that they observed during KORUS-OC that were not well-represented well by a simple Angstrom exponent and why.

2. Add a figure with a schematic of the sampling inlet, instruments connected to the sampling inlet, and flows. This is described in the text, but it would be clearer with a schematic showing the SpEx, TAP, and nephelometer with the inlet system and respective flows.

3. Section 2 could be better organized for readability. The current organization of the paper is:

Introduction: Includes description of KORUS-OC field campaign

Section 2.1: Further description of KORUS-OC field campaign, shared sampling inlet, and instrument operation. TAP instrument is introduced here, but not defined.

Section 2.2: Description of absorption (TAP) and scattering (nephelometer) instruments, calibrations, ship plume interferences, and wavelength corrections.

Section 2.3: Description of SpEx instrument, modifications since Jordan et al 2015, and data filtering.

One suggestion for alternative headings and organization:

Section 2.1: Overview of KORUS-OC field campaign, ship, and ship tracks

Section 2.2: Description of three sampling instruments, calibrations, and data processing

Section 2.3: Description of shared instrument inlet with schematic

4. It would be simpler to present all of the measurements in nm or um throughout the paper, rather than changing from nm to um in Section 3.3.

Other comments:

Line 74: Add reference to Chartier and Greenslade 2012?

Line 135: Indicate that the size cut was determined theoretically.

Line 175: What does NT stand for? "Nephelometer TAP"?

Line 242: What does "Blocking" refer to? Could this be explained?

---

## Referee Comment (RC2) · Anonymous Referee #2 · 8 Sep 2020

Title: New In Situ Aerosol Hyperspectral Optical Measurements over 300-700 nm, Part 1: Spectral Aerosol Extinction (SpEx) Instrument Field Validation during the KORUS-OC cruise

Authors: Carolyn E. Jordan, Ryan M. Stauffer, Brian T. Lamb, Charles H. Hudgin, Kenneth L. Thornhill, Gregory L. Schuster, Richard H. Moore, Ewan C. Crosbie, Edward L. Winstead, Bruce E. Anderson, Robert F. Martin, Michael A. Shoo, Luke D. Ziemba, Andreas J. Beyersdorf, Claire E. Robinson, Chelsea A. Corr and Maria A. Tzortziou

[Figure]

General Comments: This is a well written paper about an important topic: the wavelength variation of aerosol extinction. The in situ aerosol extinction data as measured by SpEx presented in this paper are convincing and an important contribution to the literature as most if not all spectral extinction data published to date are column integrated from remote sensing measurements of the total atmospheric column (AOD). However there are a few issues in the manuscript that I think should to be expanded upon or require some clarification. I think the authors should discuss in greater detail (in the text) the departure of the sampling RH from ambient by ∼25% to ∼30% (SpEx sampled particles are drier). This is significant since particle scattering due to particle growth increases exponentially as RH increases. See Kotchenruther et al. (1999; JGR), figure 5 of that paper. Therefore, light scattering at ambient RH under high humidity conditions would sometimes be significantly greater than that measured by SpEx. This also has potential implications for the measurement or computation of ambient single scattering albedo. Additionally, please provide some discussion on why plotting the data in spectral fit coefficient space (a1, a2; Figs 9 & 10) is better or more informative than utilizing the two spectral parameters of Angstrom Exponent (AE; for say 370-700 nm) and its Curvature (AE') as defined by a 2nd order fit of extinction versus wavelength (logarithmic), again over the entire measured WL range. It would seem to be more physically intuitive to most readers to analyze the data in this manner than to analyze and plot the fit coefficients. I suggest acceptance by the journal after the above and following comments/suggestions have been considered, and the manuscript revised in response.

Specific Comments:

Page 2, Lines 35-36: Perhaps this sentence should be re-written to suggest that the combination of both Angstrom Exponent and spectral curvature information provides the most information related to particle size distribution.

Page 3, Lines 65-66: I think this statement is too strong. Hyperspectral data are not really required since the wavelength dependence of extinction varies smoothly with

wavelength. Therefore sufficient wavelength sampling does not need to be hyperspectral, but does require several narrowband spectral measurements spanning the UV, visible and NIR wavelengths.

Page 3, Lines 67-74: It would be appropriate in this section to also discuss the work of O'Neill et al. (2001, 2003) that utilized the 2nd order fit to AOD spectra (parameter AE') and AE to separate fine mode versus coarse mode AOD components. This algorithm was also successfully applied by Kaku et al. (2014; AMT) to in situ air sampling instrumentation on ship to determine fine and coarse mode extinction components.

Page 4, Lines 94: FOV is typically used to define pixel size, so perhaps viewing region or something like that is more accurate in the context of this sentence.

Page 4 Line 123-124: It would be very useful to be clearer here. Did you make adjustments to bring the SpEx data close to ambient conditions or leave them as extinction spectra of partially dried aerosol?

Page 8 Line 271-272: The slope of near unity occurs in Figure 3 since it seems that both instruments have partially dried aerosol with RH lower than ambient by ∼25% to ∼30%. Perhaps this should be mentioned in the text when discussing this Figure.

Page 9 line 305: Please discuss why the data are noisier in Fig 7 at the longest wavelength end of the SpEx measured extinction coefficients. Is this consistent for all measured spectra from SpEx? Is this random variation or does it depend on temperature or some other variable?

Page 11 line 353, equation 5: It would be useful to state that -2a2 is equal to the parameter AE' which defines the curvature of extinction spectra, AE'=-dAE/d lnWL .

Page 11 line 366-367: Note that another significant difference between the SpEx extinction coefficient data and AERONET AOD data is that in the total atmospheric column measured by AERONET there is always some variable amount of coarse mode particles present, while the SpEx sampling excludes all coarse mode particles and
even the shoulder of some very large fine mode particles (see Dall'Osto et al., 2009 and Eck et al. 2018). Coarse mode (super-micron sized radius) particles have Angstrom Exponent close to zero (actually slightly negative) and also curvature near to zero. I suspect that this difference in particle size sampling is of greater importance than the spectral range since the 2nd order fit is excellent with AERONET data (within measurement uncertainty) throughout the 380 to 870 nm wavelength range. Note that the 1020 nm AOD data in the AERONET database in 2006 had greater uncertainty due to water vapor absorption in the Schuster et al. 2006 paper than current V3 data and also that the 340 nm channel has significantly larger uncertainty than the other measured wavelengths due to interference filter issues (out-of-band blocking & higher transmission degradation rate at this wavelength). For these reasons both the computation of AE' (Eck et al., 1999, 2001 etc) and SDA retrievals (O'Neill et al. 2001, 2003 etc) of fine and coarse AOD only utilize the 380, 440, 500, 675 and 870 nm wavelengths from the AERONET database. Also the 340 nm and 1020 nm are excluded from the AOD input to the SDA algorithm in the AERONET database due to significantly larger uncertainties and potential biases at those wavelengths.

Page 12 line 419-421: It seems that only full spectra (or at least encompassing the wavelength extremes and mid-point) should be analyzed. This is the strategy with making SDA retrievals from AERONET data at Level 2 (publication quality). For AERONET the 380, 500 and 870 nm wavelengths at a minimum must be available thus encompassing the minimum, maximum and middle wavelengths over the wavelength range considered. This ensures an accurate characterization of the non-linearity of the AOD spectra. The other possible wavelengths are 440 and 675 nm and are utilized in addition to the minimum three, if available.

Figure 7, x-axis labeling and caption: It would be much more useful for most readers if you label the x-axis in wavelength (with logarithmic scale) either in nm or microns not as logarithm of wavelength.

[Figure]

---

## Referee Comment (RC3) · Anonymous Referee #3 · 23 Sep 2020

This manuscript describes spectrally resolved aerosol extinction measurements taken with a White-type optical cell during KORUS-OC campaign during the summer of 2016. The performance of the system was evaluated by comparison to the sum of scattering and absorption measurements done with commercially available instruments. Perhaps the most significant issue discussed in this manuscript is the pitfall in using the extinction Angstrom exponent (EAE) to describe the spectral dependence of optical properties. The authors compare and discuss the advantages of alternatively using a 2D parameter space (namely, the two first fitting parameters of a 2nd order polynomial fit).

I recommend this manuscript be published after addressing the following comments. General comments: 1) The title of the manuscript does not seem to describe the main topic I understood it. From the title, it would seem that the field validation is the main topic while to me it seemed almost the trivial part. There is no mention in the title of the fact that the instrument was used to compare and discuss the advantages of an alternative to the EAE. 2) I believe that the manuscript could be much better organized. The text often refer the reader to plots not in the order of their appearance and to several plots at the same line. This makes it a bit less readable and difficult to follow the story. 3) The use of the 2D parameter space as an alternative to EAE (i.e. to derive information on the aerosol size distribution) is discussed only in the context of the data presented here. I believe it would strengthen the manuscript to add a more general discussion, perhaps with the aid of simulated data. 4) The reason for the large errors in fitting an EAE to spectral data, namely "larger" particles is not mentioned (aside from one sentence towards the end of the manuscript, line 417-418). The curvature aspect of the log-log data can be easily explained in terms of Mie theory for light extinction by spherical particles. The classical Qext Vs size parameter curve shows smooth increase behavior for "small" size parameters while it wiggles over "larger" size parameters due to Mie resonances. "larger" size parameters mean larger particles and/or shorter wavelengths. It is these wiggles (due to Mie resonances) that cause the apparent deviation from power law behavior. The larger the particles (and/or shorter the wavelength) the more wiggles fit into the measured spectral window and more orders are needed for a reasonable fit using a polynomial. This means that the smaller the mode diameter and the smaller the sd of the aerosols size distribution the more likely the measured log-log spectrum will be adequately represented with a linear (1st order polynomial) fit. 5) My impression was that there is a fair amount of text and some figures that are not essential to deliver the main message. removing them (maybe to the supplementary material) would some room for more needed discussion. 6) A few simple simulations that I did showed inverse (almost) linear relationship between a2 and a1. This manuscript shows a positive (almost) linear relationship (note the inverted x-axis

in figures 9 and 10. I could be mistaken but I suggest the authors take a second look to check this issue. Specific comments: Line 28: I suggest using the convention of EAE (Extinction Angstrom Exponent) rather than $\alpha$ which can be easily confused with the symbol often used to describe extinction coefficient. Line 33-35: after reading the final part of the abstract several times I still don't understand what is exactly the message here. The characteristic wavelength is not a well-known property of aerosol population and is not explained in the abstract. Therefore, to me it sounds like one would need to read the whole paper to understand this part. This is surly not something you would want for the abstract. Additionally, (1) "such that . . .aerosol size distributions with the same $\alpha$. . ." how can different size distribution of aerosols have the same $\alpha$? Isn't it true that $\alpha$ is heavily dependent on the size distribution? (2) is the slope related to $\lambda$ch or is it the $\lambda$ch? Line 81: it is not clear to me what the authors meant by "2 nm resolution in intensity" Line 117: TAP is not defined. Line 135: sampling off (from/by/using?) the same inlet line? Line 141-142: "as the ship. . . at that time". It is not clear to me what do you mean by "restarting the system". I also feel this sentence does not contribute to the manuscript. Line 148-149: the internet links are not needed and in my opinion reduce readability. Line 165: here, the reader is asked to compare figure S3 and figure 2 but there is no explanation as to what is the difference between the two figures. It should also be made clear that this data was acquired by using the TAP and IN. Line 174: a discussion about the errors generated from the wavelength adjustment is needed. Line 263-265: how were local ship emissions distinguished from emissions of the research vessel? Line 272: the plot referring to the 632nm channel shows that all fitted lines consistently show 5-9% over estimation of extinction from sum of scattering and absorption compared with measured extinction. What could be the cause? How accurate are the absorption measurements in this wavelength? Is it possible that these have a positive bios? Line 280-281: it is not clear to me what is meant here by "10 s scat and 1 s abs measurement for every ext spectrum. Please explain. Line 303: it would be beneficial to choose one unit system (wavelength in nm or um) and b consistent throughout the manuscript. Line 350-355: this is an important paragraph which

I strongly feel that is not explained properly. (1) "The two expressions used here to fit the relationship between $\sigma$ext and $\lambda$ are related by their negative derivative, defined as $\alpha$ in Eq. (2)". Eq. 2 applies by definition only to the linear fit so how does it relate the linear fit to the 2nd order poly' fit? (2) "the derivative of the linear fit (y=a+bx; dy/dx = b) equals the derivative of the 2nd order polynomial fit (y = a0 +a1x+a2x2; dy/dx = a1 +2a2x) such that…" this statement seems wrong to me. It implies that for every $\lambda$ the expression (a1+2a2*ln($\lambda$)) has the same value. Did you mean to write that Eq 5 is only valid for one $\lambda$? That is $\lambda$ch? Line 357: "The wavelength dependence of Eq. (5)" if I understand correctly, Eq. 5 is not dependent on wavelength (i.e. the spectra of the measured data, 300-700 nm). It is dependent on the characteristic wavelength ($\lambda$ch) as it is defined in the lines above. Line 360: did you mean figure 9? Not sure how figure 10 is related here. Line 360-362: it is clear that when a2 = 0 the 2nd order poly' fit is actually a line fit. What is the physical interpretation of $\lambda$ch = 1 um? Line 364: what does it mean? Line 369: to the right and left of which distribution? Lines 376-378: "but as discussed…space". This section is realy not clear. If the intension here was to link the information about the sampled aerosols that is available in other publications and that is presented in section 3.1 to the fitted parameter space, this is was not made clear. Please rephrase. Figure 4 top: the gray trace is not visible. Figure 6: it would be more intuitive for me to plot these data sets with the EAE calculated from the full range on x axis because this represents the optimal case and demonstrates the variances is most of the spectra to that case. Figure 7: what information is presented or made clearer in the top two figures that is not presented or is not clear in the middle two figures? In my opinion there is no add value to the top two and therefore should be removed. Additionally, the residual could be presented in relative terms (i.e. % error). This would be more intuitive to understand and reduce the text needed to describe the fit errors. Figure 8: here I would also suggest showing the x-axis in terms of relative error. Figure 9: in this figure all x-axis are inverted. This makes the figure less intuitive and harder to understand. Is there a reason for this choice? If yes I believe it should be explain in the text or at least in the figure caption. What is the purpose of the rectangle

inset in the bottom panel of figure 9? Figure S2: it is not clear what this data set is based on. Is it simulation or measurements? If data is from another reference which is it? Figure S3: the full range of the right axis is not needed. Values range from 0 to 2. Figure S4: what do the authors mean by hour set of intensity spectra? Is it a one hour average?
* * *

---

## Author Comment (AC1) · 20 Nov 2020

***Response to referee comments on*** "New In Situ Aerosol Hyperspectral Optical Measurements over 300–700 nm, Part 1: Spectral Aerosol Extinction (SpEx) Instrument Field Validation during the KORUS-OC cruise" ***by*** **Carolyn E. Jordan et al.**

On behalf of my co-authors and myself, we thank the referees for the time and effort they made in their careful review of this work. We very much appreciate their insights and suggestions which have led to revisions that we think substantively improve this manuscript. In our responses below we use gray italics for the comments from the referees, black regular font in our responses to those comments, and black italicized font for quoted material from the revision. Where we refer to line numbers, the numbers from the submitted manuscript are used in order to be consistent with the referees' comments (with the new numbers from the revision enclosed in parentheses). Track changes in the revised manuscript also make it straightforward to find all of the revised text.

**Anonymous Referee #1**

*This paper describes the first field measurements by the spectral aerosol extinction (SpEx) instrument. The SpEx instrument measures broadband aerosol extinction at 300 - 700 nm with 0.8 nm resolution. It was deployed on a ship cruise around the Korean peninsula during May – June 2016.*

*The paper presents Angstrom exponents and second-order polynomial fits to the extinction data, and determines that second-order polynomial fits are a better representation.*

*This paper builds on previous instrument development work described in Jordan et al 2015 AMT and Chartier and Greenslade 2012 AMT.*

*Major comments:*

*1. The manuscript would be improved by examining the underlying relationship between aerosol extinction and wavelength.*

*As described in the introduction, the major intrinsic properties that determine aerosol extinction are the aerosol size distribution, aerosol shape, and complex refractive index. The Angstrom exponent, A (where Extinction = k wavelengthˆ-A) is an empirical expression to describe the wavelength-dependence of the aerosol extinction. Some combinations of size distribution, shape, and complex refractive index produce extinction values that are not well-represented by an Angstrom exponent.*

*This paper presents Angstrom exponents and an alternative second-order polynomial fit to the extinction data. The paper would be much stronger if the authors calculated the expected aerosol extinction from the size distribution, assumed shape, and complex refractive index (volume-weighted by composition). They could then describe the types of aerosol populations that they observed during KORUS-OC that were not well-represented well by a simple Angstrom exponent and why.*

We sincerely appreciate this suggestion and share the referee's interest in conducting this type of study.  Unfortunately, we do not have size distribution data measured aboard the *R/V Onnuri* that we can directly compare to the optical measurements that were made.  To the extent that we use the published record from KORUS-AQ to provide the overarching regional context for our measurements, that is not the same as having measurements for direct comparison.  We have more recently acquired data from a subsequent field campaign where we do have commensurate size distribution and composition data and we plan to undertake the kind of study the referee recommends with that more complete data set.

*2. Add a figure with a schematic of the sampling inlet, instruments connected to the sampling inlet, and flows. This is described in the text, but it would be clearer with a schematic showing the SpEx, TAP, and nephelometer with the inlet system and respective flows.*

This has been added to the supplement (Fig. S2) cited in the second sentence of Section 2.1 (line 114 (146)).  Note, Fig. S1 and S2 in the following sentence are not out of order as Fig. S1 is cited in the Introduction:

*"The instrument suite (Fig. S2) was deployed above the bridge strapped to the starboard rail in a custom-built box designed to keep the instruments dry yet ventilated to prevent overheating (Fig. S1)."*

*3. Section 2 could be better organized for readability. The current organization of the paper is:*

*Introduction: Includes description of KORUS-OC field campaign*

*Section 2.1: Further description of KORUS-OC field campaign, shared sampling inlet, and instrument operation. TAP instrument is introduced here, but not defined.*

*Section 2.2: Description of absorption (TAP) and scattering (nephelometer) instruments, calibrations, ship plume interferences, and wavelength corrections.*

*Section 2.3: Description of SpEx instrument, modifications since Jordan et al 2015, and data filtering.*

*One suggestion for alternative headings and organization:*
*Section 2.1: Overview of KORUS-OC field campaign, ship, and ship tracks*

*Section 2.2: Description of three sampling instruments, calibrations, and data processing*

*Section 2.3: Description of shared instrument inlet with schematic*

We have modified the text to introduce the IN101 and TAP by name earlier.  At the end of the Introduction the sentence starting at line 102 (132) now reads:

*"Two commercial instruments (AirPhoton's integrating nephelometer, IN101, and Brechtel's Tricolor Absorption Photometer, TAP) were also deployed to measure in situ aerosols at three visible wavelengths providing scattering coefficients and absorption coefficients, respectively."*

We then revised the first paragraph of Section 2.2 and the first sentence of the next paragraph as follows:

*"The IN101 and TAP instruments provide data at a higher temporal resolution than SpEx and were deployed with two objectives:  1) to identify and flag incidents of ship exhaust ("plume") contamination of the data set, and 2) to evaluate the new spectral measurements (both from SpEx and the filters).  The TAP (model 2901, Brechtel, Hayward, CA) measures absorption coefficients ($\sigma_{abs}$) at 467, 528, and 652 nm with 1 s resolution and the IN101 (AirPhoton, Baltimore, MD) measures scattering coefficients ($\sigma_{scat}$) at 450, 532, and 632 nm with ~10 s resolution.*

*To identify ship plume interceptions, an initial examination of the IN101 $\sigma_{scat}$ and TAP $\sigma_{abs}$ data was performed."*

We think these changes resolve some of the out-of-order problems that Referee #3 also noted, while preserving balance across the 3 subsections (3 paragraphs in Section 2.1, 4 paragraphs each for Sections 2.2 and 2.3).

*4. It would be simpler to present all of the measurements in nm or um throughout the paper, rather than changing from nm to um in Section 3.3.*

In our previously published work (Jordan et al., 2015) and various presentations at meetings we have consistently used nm to refer to the wavelength range of the SpEx instrument.  In general, it is more convenient to use integers rather than fractional values.  The switch to μm in Section 3.3 arises from the need to use units consistent with those from the Schuster et al. (2006) publication in order to compare our results to theirs.  In response to comments from Referees #2 and #3 we have revised the beginning of Section 4 to include more detail on our nomenclature and the implications of Eq. (5).  This includes a better explanation on the necessity to be cognizant of the units when calculating spectral curvature coefficients.  We think this is an important point to make, since the calculation of $\alpha_{ext}$ is wavelength independent and those who use it to extrapolate measurements from a few visible wavelengths may not be aware that using nm rather than μm for curvature calculations is problematic.  For this reason, keeping both sets of units in this work is more informative.

*Other comments:*

*Line 74: Add reference to Chartier and Greenslade 2012?*

We thank the referee for catching this oversight!  To give Meg Greenslade the credit she is due, rather than adding a citation at line 74 we changed the first sentence in Section 2.3 to:

*"Developed from a prototype described in Chartier and Greenslade (2012) SpEx is described in detail in Jordan et al. (2015)."*

*Line 135: Indicate that the size cut was determined theoretically.*

We have updated the figure caption (note the new number due to the addition of a new Fig. S2) in the supplement to state:

**"Figure S3.**  *Illustration of the relationship between flow rate and size cut of the sampled aerosol particles based on theoretical calculations pertinent to the system deployed aboard the R/V Onnuri."*

*Line 175: What does NT stand for? "Nephelometer TAP"?*

The sentence on line 175 (232) now reads:

*"For clarity, these extinction coefficients are denoted NT (for Nephelometer + TAP) $\sigma_{ext}$ in comparison to the measured SpEx $\sigma_{ext}$ at those wavelengths (450, 532, and 632 nm)."*

*Line 242: What does "Blocking" refer to? Could this be explained?*

We have added a description of the Rex Block that gives the Blocking period its name in the sentence starting at line 244 (311) and broken the original sentence into 2 parts:

*"The Blocking period was then characterized by limited transport and occasional brief stagnant periods due to adjacent high and low pressure systems with the high poleward of the low (called a Rex Block).  Under these conditions local sources dominated pollutants, but aerosols did not accumulate to large concentrations (Peterson et al., 2019; Jordan et al., 2020a)."*

**Anonymous Referee #2**

*General Comments: This is a well written paper about an important topic: the wavelength variation of aerosol extinction. The in situ aerosol extinction data as measured by SpEx presented in this paper are convincing and an important contribution to the literature as most if not all spectral extinction data published to date are column integrated from remote sensing measurements of the total atmospheric column (AOD). However there are a few issues in the manuscript that I think should to be expanded upon or require some clarification. I think the authors should discuss in greater detail (in the text) the departure of the sampling RH from ambient by ~25% to ~30% (SpEx sampled particles are drier). This is significant since particle scattering due to particle growth increases exponentially as RH increases. See Kotchenruther et al. (1999; JGR), figure 5 of that paper. Therefore, light scattering at ambient RH under high humidity conditions would sometimes be significantly greater than that measured by SpEx. This also has potential implications for the measurement or computation of ambient single scattering albedo. Additionally, please provide some discussion on why plotting the data in spectral fit coefficient space (a1, a2; Figs 9 & 10) is better or more informative than utilizing the two spectral parameters of Angstrom Exponent (AE; for say 370-700 nm) and its Curvature (AE') as defined by a 2nd order fit of extinction versus wavelength (logarithmic), again over the entire*

*measured WL range. It would seem to be more physically intuitive to most readers to analyze the data in this manner than to analyze and plot the fit coefficients. I suggest acceptance by the journal after the above and following comments/suggestions have been considered, and the manuscript revised in response.*

The point about RH is very well taken. We have revised the text to clarify our approach, specifically in response to the comment about Page 4 Line 123-124 below, which we copy here for your convenience:

We have clarified the text earlier in this paragraph (line 149 of revised manuscript) to explain that when we say measurements were at ambient, we mean that the aerosols were not dried prior to sampling. We then updated the conclusion of the paragraph as follows:

*"The primary objective in this work is to evaluate the performance of SpEx by direct comparisons to the data from the two commercial instruments in the measurement suite. Hence, we did not perform any corrections to either ambient T and RH or to standard temperature and pressure (STP) prior to comparing these data. There are no comparisons in this pair of manuscripts to other data sets, so such corrections are not necessary for this study."*

As for the referee's second point above, as detailed in our responses to the specific comments below, we have more fully described the wavelength dependence of the curvature coefficients in Section 4 of the revised manuscript, including a much better explanation of the importance of the characteristic wavelength ($\lambda_{ch}$) of the measurement. We now summarize that expanded discussion in the abstract (starting at line 31 of both versions of the manuscript) as follows:

*"Building on previous studies that used total column AOD observations to examine the information content of spectral curvature, the relationship between $\alpha$ and the second order polynomial fit coefficients ($a_1$ and $a_2$) was found to depend on the wavelength range of the spectral measurement such that any given $\alpha$ maps into a line in ($a_1$,$a_2$) coefficient space with a slope of $-2ln(\lambda_{ch})$, where $\lambda_{ch}$ is defined as the single wavelength that characterizes the wavelength range of the measured spectrum (i.e., the "characteristic wavelength"). Since the curvature coefficient values depend on $\lambda_{ch}$, it must be taken into account when comparing values from spectra obtained from measurement techniques with different $\lambda_{ch}$. Previously published work has shown that different bimodal size distributions of aerosols can exhibit the same $\alpha$, yet have differing spectral curvature with different ($a_1$,$a_2$). This implies that ($a_1$,$a_2$) contain more information about size distributions than $\alpha$ alone. Aerosol size distributions were not measured during KORUS-OC and the data reported here were limited to the fine fraction, but the ($a_1$,$a_2$) maps obtained from the SpEx data set are consistent with the expectation that ($a_1$,$a_2$) may contain more information than $\alpha$, a result that will be explored further with future SpEx and size distribution data sets."*

*Specific Comments:*

*Page 2, Lines 35-36: Perhaps this sentence should be re-written to suggest that the combination of both Angstrom Exponent and spectral curvature information provides the most information related to particle size distribution.*

We have revised this part of the abstract (starting at line 31 of both versions of the manuscript, please see our response to your overall comments above).

*Page 3, Lines 65-66: I think this statement is too strong. Hyperspectral data are not really required since the wavelength dependence of extinction varies smoothly with wavelength. Therefore sufficient wavelength sampling does not need to be hyperspectral, but does require several narrowband spectral measurements spanning the UV, visible and NIR wavelengths.*

We have revised the sentence on lines 65-66 (82-84) as follows:

*"However, if $\alpha$ is not, in fact, wavelength-independent for ambient aerosols, then a hyperspectal measurement (or sufficient wavelength sampling spanning the full wavelength range of interest) is required to capture the actual wavelength dependence."*

*Page 3, Lines 67-74: It would be appropriate in this section to also discuss the work of O'Neill et al. (2001, 2003) that utilized the 2nd order fit to AOD spectra (parameter AE') and AE to separate fine mode versus coarse mode AOD components. This algorithm was also successfully applied by Kaku et al. (2014; AMT) to in situ air sampling instrumentation on ship to determine fine and coarse mode extinction components.*

Thank you for this suggestion. The intent of that paragraph was focused on the influence of aerosol microphysical and chemical properties on spectral shape and how SpEx can relate the latter to other in situ observations of the former. However, we had not thought to also mention how it can provide a new link between in situ observations and retrievals. We have revised the end of that paragraph (starting at line 92 of the revised manuscript) as follows:

*"Recently, the retrieval algorithm developed by O'Neill et al. (2001, 2003, 2008) to distinguish fine mode from coarse mode AOD components using $2^{nd}$ order spectral fits has been applied to in situ extinction components based on scattering and absorption measurements made at a few visible wavelengths and evaluated using several sets of field data (Kaku et al., 2014, and references therein). The Spectral Aerosol Extinction (SpEx) instrument (Jordan et al., 2015) provides a new measurement approach that combines the advantages of a broad spectral range (typically limited to remote sensing techniques) with an in situ measurement capability (that allows for direct comparison to other in situ measurements of ambient aerosol microphysical and chemical properties). Combining this hyperspectral in situ measurement capability with retrieval techniques as in Kaku et al. (2014) also provides a new tool to fine tune remote sensing retrievals."*

We then relocated the text citing the spectral resolution to the next paragraph and updated the conclusion of that paragraph to again emphasize the ability to use SpEx to relate in situ spectral extinction to aerosol microphysics and composition as follows:

*"Hence, SpEx is particularly suited to examine spectral details for ambient aerosols over its measurement range and to relate those spectral details to simultaneous in situ measurements of ambient aerosol microphysics and composition."*

*Page 4, Lines 94: FOV is typically used to define pixel size, so perhaps viewing region or something like that is more accurate in the context of this sentence.*

We have changed this to field of regard (line 123 of the revised manuscript, field of view was used by mistake, thanks for catching this!).

*Page 4 Line 123-124: It would be very useful to be clearer here. Did you make adjustments to bring the SpEx data close to ambient conditions or leave them as extinction spectra of partially dried aerosol?*

We have clarified the text earlier in this paragraph (line 149 of revised manuscript) to explain that when we say measurements were at ambient, we mean that the aerosols were not dried prior to sampling.  We then updated the conclusion of the paragraph as follows:

*"The primary objective in this work is to evaluate the performance of SpEx by direct comparisons to the data from the two commercial instruments in the measurement suite.  Hence, we did not perform any corrections to either ambient T and RH or to standard temperature and pressure (STP) prior to comparing these data.  There are no comparisons in this pair of manuscripts to other data sets, so such corrections are not necessary for this study."*

*Page 8 Line 271-272: The slope of near unity occurs in Figure 3 since it seems that both instruments have partially dried aerosol with RH lower than ambient by ~25% to ~30%. Perhaps this should be mentioned in the text when discussing this Figure.*

We think that with the revised text in section 2.1 (shown in our response to your recommendation above) accomplishes this.

*Page 9 line 305: Please discuss why the data are noisier in Fig 7 at the longest wavelength end of the SpEx measured extinction coefficients. Is this consistent for all measured spectra from SpEx? Is this random variation or does it depend on temperature or some other variable?*

This is partly an optical illusion due to the log scale used to plot the spectra.  Nonetheless, the calculation of $\sigma_{ext}$ (Eq. 4) is sensitive to the difference in the sample and reference spectra such smaller reference values ($I_0$) combined with small differences between I and $I_0$ lead to greater uncertainty in $\sigma_{ext}$.  We have added the following text at the end of the paragraph you reference (starting at line 392 of the revised manuscript) to explain this as follows:

*"Note, the log scale used to plot the spectrum (red curves) in Fig. 7 along with the relatively small extinctions at long wavelengths exaggerates the appearance of noise at those wavelengths (i.e., ± 5 Mm$^{-1}$ at the red end where $\sigma_{ext}$ ~ 20 Mm$^{-1}$, LN(20) ~ 3, is more obvious than in the UV where $\sigma_{ext}$ ~ 150 Mm$^{-1}$, LN(150) ~ 5).  Nonetheless, the intensity of the xenon lamp decreases from 600 to 700 nm (Fig. S4; in Fig. 7, LN(0.6 μm) ~ -0.51, LN(0.7 μm) ~ -0.36) such that smaller values of $I_0$ combined with the small differences between I and $I_0$ in this wavelength range lead to slightly greater uncertainty in $\sigma_{ext}$ (Eq. (4))."*

*Page 11 line 353, equation 5: It would be useful to state that -2a2 is equal to the parameter AE'
which defines the curvature of extinction spectra, AE'=-dAE/d lnWL .*

Following Eq. (5) we have added the following sentence:

*"Note, the derivative of Eq. (5), $\alpha_{ext}' = - d\alpha_{ext}/ dln(\lambda) = -2a_2$ defines the curvature of the extinction spectra (Eck et al., 1999)."*

*Page 11 line 366-367: Note that another significant difference between the SpEx extinction
coefficient data and AERONET AOD data is that in the total atmospheric column measured by
AERONET there is always some variable amount of coarse mode particles present, while the
SpEx sampling excludes all coarse mode particles and even the shoulder of some very large fine
mode particles (see Dall'Osto et al., 2009 and Eck et al. 2018). Coarse mode (super-micron
sized radius) particles have Angstrom Exponent close to zero (actually slightly negative) and
also curvature near to zero. I suspect that this difference in particle size sampling is of greater
importance than the spectral range since the 2nd order fit is excellent with AERONET data
(within measurement uncertainty) throughout the 380 to 870 nm wavelength range. Note that the
1020 nm AOD data in the AERONET database in 2006 had greater uncertainty due to water
vapor absorption in the Schuster et al. 2006 paper than current V3 data and also that the 340 nm
channel has significantly larger uncertainty than the other measured wavelengths due to
interference filter issues (out-of-band blocking & higher transmission degradation rate at this
wavelength). For these reasons both the computation of AE' (Eck et al., 1999, 2001 etc) and
SDA retrievals (O'Neill et al. 2001, 2003 etc) of fine and coarse AOD only utilize the 380, 440,
500, 675 and 870 nm wavelengths from the AERONET database. Also the 340 nm and 1020 nm
are excluded from the AOD input to the SDA algorithm in the AERONET database due to
significantly larger uncertainties and potential biases at those wavelengths.*

We clearly failed to adequately explain $\lambda_{ch}$ in the submitted version.  Based on your comments
and those of Referee #3 we have substantially rewritten Section 4.  We have clarified at the start
of the section (line 431 of the revised manuscript) that we only compare our results to the
appropriate subset of data in Schuster et al. (2006):

*"A comparison of the coefficients obtained from SpEx to the fine fraction subset of aerosols
reported in Schuster et al. (2006) revealed two key differences between the data sets."*

We have revised the paragraph following Eq. (5), starting at line 441 of the revised manuscript,
and added two new paragraphs to better explain how differing values of $\lambda_{ch}$ lead to rotation in
$(a_1,a_2)$ space (note, we include Fig. S8 below for your convenience):

*" Note, the derivative of Eq. (5), $\alpha_{ext}' = - d\alpha_{ext}/ dln(\lambda) = -2a_2$ defines the curvature of the extinction
spectra (Eck et al., 1999).  For any given spectrum, there is one wavelength at which the linear
and 2nd order polynomial fits yield equivalent results in Eq. (5).  This must not be confused with
every wavelength measured in the spectrum, so we will refer to this one wavelength as the
characteristic wavelength of the measurement range, $\lambda_{ch}$, from here on.  It can be calculated for
each measured spectrum from the two sets of fit coefficients for that spectrum.  That is, rewriting
Eq. (5) in terms of $\lambda_{ch}$ the characteristic wavelength of the measured spectrum may be calculated*

*from $\lambda_{ch} = e^{\wedge}((\alpha_{ext} + a_1) / -2a_2)$.  For the SpEx data set, $\lambda_{ch}$ was found to range from 0.36 - 0.46 μm.  In contrast, the empirical fit of $\alpha_{ext} = a_2 - a_1$ implies $\lambda_{ch} \sim 0.61$, i.e., ln(0.61) $\sim$ -0.5.  The dependence of Eq. (5) on the characteristic wavelength, results in spectra sets with differing $\lambda_{ch}$ exhibiting different mapping between $\alpha_{ext}$ and $(a_1, a_2)$.  To illustrate this, consider the range of $\alpha_{ext}$ values (0.29 - 3.25) found from linear fits over 0.3 - 0.7 μm to all of the spectra measured by SpEx during KORUS-OC.  This range of $\alpha_{ext}$ values maps differently into $(a_1, a_2)$ space as a function of $\lambda_{ch}$ (Fig. 9, top right panel).*

*There are two special cases evident in Eq. (5) that result in $\alpha_{ext} = a_1$.  First, when there is no curvature $(a_2 = 0)$, $a_1$ describes the same linear fit as $\alpha_{ext}$.  Second, when $\lambda_{ch} = 1$ μm (i.e., ln(1) = 0) Eq. (5) is insensitive to curvature such that $a_2$ can have any value at all.  This can be understood from Eq. (1), where $\alpha$ can be any value when $\lambda = 1$ μm and $p(1\mu m)$ will always $= \beta$.  The former leads to all $\lambda_{ch}$ sets overlapping at $a_2 = 0$, while the latter exhibits a broad vertical band independent of curvature $(a_2)$ (Fig. 9, top right panel).  These special cases have important implications.  As $\lambda_{ch}$ approaches 1 μm the measurement becomes insensitive to curvature, while at the short wavelengths of light represented by $\lambda_{ch} < 0.1$ μm the curvature itself becomes unimportant.  Hence, to probe spectral curvature the upper right panel of Fig. 9 shows measurement techniques with $\lambda_{ch} \sim 0.5 \pm 0.2$ μm provide the greatest sensitivity with sufficient separation in $(a_1, a_2)$ to distinguish aerosol microphysical and chemical properties influencing the spectral shape.*

*The rotation as a function of $\lambda_{ch}$ shown in Fig. 9 also illustrates why wavelength units of μm must be used to calculate $(a_1, a_2)$.  As $\lambda_{ch}$ increases to values > 1 μm, the $\alpha$ map rotates clockwise (Fig. S8).  If one used $\lambda_{ch} = 410$ nm rather than 0.41 μm, it would map into a narrow band in the next quadrant of $(a_1, a_2)$ space spanning a wide range in $a_1$ but a narrow range in $a_2$, resulting in little curvature sensitivity.  The calculation of $\alpha_{ext}$ is wavelength independent and will produce the same result no matter what units are used.  This is not the case for the calculation of $(a_1, a_2)$, so it must be emphasized that for curvature, the units matter."*

[Figure]

**"*Figure S8.* Illustration of the rotation of the mapping of α in (a₁,a₂) space as a function of λ_{ch}**
*for long wavelengths.  This illustration includes the extreme values of λ_{ch} of 100 and 1000 μm to*
*stand in for the mapping that arises from using nm units instead of μm of wavelength."*

We anticipate that when we have SpEx data representative of coarse (or bimodal size
distributions) that the (a₁,a₂) coefficients obtained from those data sets will also be rotated from
comparable size distributions in Schuster et al. (2006) due to the shorter λ_{ch} of the SpEx
measurement.

*Page 12 line 419-421: It seems that only full spectra (or at least encompassing the wavelength*
*extremes and mid-point) should be analyzed. This is the strategy with making SDA retrievals*
*from AERONET data at Level 2 (publication quality). For AERONET the 380, 500 and 870 nm*
*wavelengths at a minimum must be available thus encompassing the minimum, maximum and*
*middle wavelengths over the wavelength range considered. This ensures an accurate*
*characterization of the non-linearity of the AOD spectra. The other possible wavelengths are 440*
*and 675 nm and are utilized in addition to the minimum three, if available.*

This is an important point but bear in mind that there are different potential uses of the SpEx
data.  One would not want to use the partial spectra for a retrieval, but the above detection
portion of the spectrum might still be used in combination with absorption measurements (as in
Part 2) to look at the structure of single scattering albedo in the UV even in cases where ambient
concentrations at longer wavelengths in our instrument are below detection.  We have added the
following text to the end of that paragraph (starting at line 583 of the revised manuscript):

*"Note, partial spectra are not suitable for retrievals (i.e., comparable to those from AERONET*
*Level 2 data where at a minimum above detection values must be available from at least the*
*0.38, 0.50, and 0.87 μm channels to ensure nonlinearity in the spectrum is adequately*
*represented).  However, partial spectra can be valuable for other analyses such as when*

*combined with absorption coefficients in the calculation of ω(λ) to look for structure in the above detection range for SpEx, particularly in the UV (see Part 2, Jordan et al., 2020b). Hence, partial spectra data are not discarded from further examination."*

*Figure 7, x-axis labeling and caption: It would be much more useful for most readers if you label the x-axis in wavelength (with logarithmic scale) either in nm or microns not as logarithm of wavelength.*

The reason we use the logarithmic units on the axes of Fig. 7 is the values of the coefficients depend on those units, specifically on the natural log of wavelength in microns (as well as the natural log of $\sigma_{ext}$ in Mm$^{-1}$). Only $\alpha$ is independent of the units as is now fully explained in the text (please see above response regarding revisions to Section 4). However, to aid the reader we have included the wavelength values in µm in the figure caption that correspond to the LN(λ(µm)) x-axis labels. We have also clarified what the linear fit coefficients in Fig. 7 (a, b) represent in terms of $\beta$ and $\alpha$:

*"**Figure 7.** Example of wavelength dependence of LN($\sigma_{ext}$ (Mm$^{-1}$)) spectra (red curves) as a function of LN(wavelength (µm)). Linear (y = a + b(x), left panels; here the intercept a = LN($\beta$), the value of LN($\sigma_{ext}$) at 1 µm where LN(1 µm) = 0, and the slope b = -$\alpha$) and 2$^{nd}$ order polynomial (y = $a_0$ + $a_1$(x) + $a_2$($x^2$), right panels) fits (black curves) are shown with the fit residuals (= LN($\sigma_{ext}$ (Mm$^{-1}$)) - fit, blue curves). Residuals randomly distributed around zero indicate a good fit by the mathematical function used to fit the data, trends in residuals suggest another function may provide a better fit. Top and bottom panels show fits to a subrange (LN(0.450 - 0.632 µm)) and full range (LN(0.3 - 0.7 µm) of the measured spectrum, respectively. Middle panels show the extrapolation of the fit in the top panels over the full measured wavelength range. The x-axis labels of -1.2, -1.0, -0.8, -0.6 and -0.4 for LN(λ (µm)) equal 0.301, 0.368, 0.449, 0.549, 0.670, and 0.698 um wavelengths, respectively."*

**Anonymous Referee #3**

*This manuscript describes spectrally resolved aerosol extinction measurements taken with a White-type optical cell during KORUS-OC campaign during the summer of 2016. The performance of the system was evaluated by comparison to the sum of scattering and absorption measurements done with commercially available instruments. Perhaps the most significant issue discussed in this manuscript is the pitfall in using the extinction Angstrom exponent (EAE) to describe the spectral dependence of optical properties. The authors compare and discuss the advantages of alternatively using a 2D parameter space (namely, the two first fitting parameters of a 2nd order polynomial fit).*

*I recommend this manuscript be published after addressing the following comments.*

*General comments:*

*1) The title of the manuscript does not seem to describe the main topic I understood it. From the title, it would seem that the field validation is the main topic while to me it seemed almost the trivial part. There is no mention in the title of the fact that the instrument was used to compare and discuss the advantages of an alternative to the EAE.*

This is an interesting observation. We don't think changing the title is an option at this point (the editor can correct us if we are wrong about that). For us, the primary objective of this paper is to publish a methods paper that describes the instrument in detail that we can cite in future work and that demonstrates that the instrument works well. The secondary objective is to preemptively address a comment/question that the lead author has gotten every time this work has been presented that takes some form of "why is this useful when we can just calculate the spectrum using Angstrom exponents?". Part of the reason that it took so long to prepare this manuscript for publication was in trying to understand why the spectra were not linear as expected, then upon discovering the rich literature on spectral curvature from the remote sensing retrieval community, why the curvature coefficients from SpEx did not agree with those in Schuster et al. (2006). And so, the manuscript over time developed two somewhat distinct parts: validating the method with the commercial instruments and applying the Schuster et al. (2006) analytical approach to the data (both here and in Part 2). We anticipate a variety of analytical approaches one could take with the SpEx measurements, depending on the subject of inquiry. If the editor agrees that the title should be amended, we would be willing to consider alternatives.

*2) I believe that the manuscript could be much better organized. The text often refer the reader to plots not in the order of their appearance and to several plots at the same line. This makes it a bit less readable and difficult to follow the story.*

We checked the manuscript and the figures are introduced in order. We do make comparisons across figures, so perhaps referring back to a previously introduced figure has caused some confusion. The other concern here may be related to our choice of showing the 532 nm comparisons in the main text yet including the 450 and 632 nm comparisons in the supplement (this accounts for the times we point to Fig. 4, S6, and S7, for example). The reader does not need to refer to the figures in the supplement. The main figures are sufficient. Nonetheless, for the purpose of validating the performance of the instrument we wanted keep the other two wavelengths in the record.

*3) The use of the 2D parameter space as an alternative to EAE (i.e. to derive information on the aerosol size distribution) is discussed only in the context of the data presented here. I believe it would strengthen the manuscript to add a more general discussion, perhaps with the aid of simulated data.*

For this we recommend Schuster et al. (2006).

*4) The reason for the large errors in fitting an EAE to spectral data, namely "larger" particles is not mentioned (aside from one sentence towards the end of the manuscript, line 417-418). The curvature aspect of the log-log data can be easily explained in terms of Mie theory for light extinction by spherical particles. The classical Qext Vs size parameter curve shows smooth increase behavior for "small" size parameters while it wiggles over "larger" size parameters*

*due to Mie resonances. "larger" size parameters mean larger particles and/or shorter wave-lengths. It is these wiggles (due to Mie resonances) that cause the apparent deviation from power law behavior. The larger the particles (and/or shorter the wavelength) the more wiggles fit into the measured spectral window and more orders are needed for a reasonable fit using a polynomial. This means that the smaller the mode diameter and the smaller the sd of the aerosols size distribution the more likely the measured log-log spectrum will be adequately represented with a linear (1st order polynomial) fit.*

Thank you for the very interesting observations. Since we have not done Mie calculations as a part of this work, we have not discussed the results in these terms. As mentioned previously both in the manuscript and in this response to Referee #1, we did not measure size distributions (SD) aboard the *R/V Onnuri*. Thus, we cannot use measured SDs to provide insight, either. We anticipate a future publication using more recent data where we have both SpEx data and size distributions to give us an opportunity to relate our observations to Mie theory calculations and present such a discussion.

*5) My impression was that there is a fair amount of text and some figures that are not essential to deliver the main message. removing them (maybe to the supplementary material) would some room for more needed discussion.*

Thank you. We hope that the reviewer finds a more focused document in the latest manuscript.

*6) A few simple simulations that I did showed inverse (almost) linear relationship between a2 and a1. This manuscript shows a positive (almost) linear relationship (note the inverted x-axis in figures 9 and 10. I could be mistaken but I suggest the authors take a second look to check this issue.*

Thank you. We did check again and obtained the same result. Did you use nm or μm? The choice in units matters here as discussed below and in greater detail in the revised text.

*Specific comments:*

*Line 28: I suggest using the convention of EAE (Extinction Angstrom Exponent) rather than α which can be easily confused with the symbol often used to describe extinction coefficient.*

Too many redundant symbols for common parameters is always a problem in our business. We have adopted a nomenclature herein to avoid confusion with the use of AE / AE'. We chose the Greek symbol α because it is commonly used to represent the extinction Angstrom Exponent in the literature.

*Line 33-35: after reading the final part of the abstract several times I still don't understand what is exactly the message here. The characteristic wavelength is not a well-known property of aerosol population and is not explained in the abstract. Therefore, to me it sounds like one would need to read the whole paper to understand this part. This is surly not something you would want for the abstract. Additionally, (1) "such that . . .aerosol size distributions with the same α. . ."*

*how can different size distribution of aerosols have the same α? Isn't it true that α is heavily dependent on the size distribution? (2) is the slope related to λch or is it the λch?*

This is a very good point. The characteristic wavelength is a term we define (for the first time, we believe) in this work. And as you will see below, we have a more extensive discussion of this in the revised text of Section 4. Here, we have updated the abstract (starting at line 31) to more fully explain both $\lambda_{ch}$ and the idea behind the greater information content of $(a_1, a_2)$ as follows:

*"Building on previous studies that used total column AOD observations to examine the information content of spectral curvature, the relationship between α and the second order polynomial fit coefficients ($a_1$ and $a_2$) was found to depend on the wavelength range of the spectral measurement such that any given α maps into a line in ($a_1, a_2$) coefficient space with a slope of $-2\ln\lambda_{ch}$, where $\lambda_{ch}$ is defined as the single wavelength that characterizes the wavelength range of the measured spectrum (i.e., the "characteristic wavelength"). Since the curvature coefficient values depend on $\lambda_{ch}$, it must be taken into account when comparing values from spectra obtained from measurement techniques with different $\lambda_{ch}$. Previously published work has shown that different bimodal size distributions of aerosols can exhibit the same α, yet have differing spectral curvature with different ($a_1, a_2$). This implies that ($a_1, a_2$) contain more information about size distributions than α alone. Aerosol size distributions were not measured during KORUS-OC and the data reported here were limited to the fine fraction, but the ($a_1, a_2$) maps obtained from the SpEx data set are consistent with the expectation that ($a_1, a_2$) may contain more information than α, a result that will be explored further with future SpEx and size distribution data sets."*

*Line 81: it is not clear to me what the authors meant by "2 nm resolution in intensity"*

The resolution of polarimeters is described both in terms of the spectral resolution for intensity and for the degree and angle of linear polarization. For the sake of the comparisons in this paragraph the resolution in intensity is the parameter of interest.

*Line 117: TAP is not defined.*

In response to Referee #1's request to improve the readability of Section 2 we now introduce the TAP (and IN101) at the end of the Introduction (starting at line 102 (132)), resolving this problem:

*"Two commercial instruments (AirPhoton's integrating nephelometer, IN101, and Brechtel's Tricolor Absorption Photometer, TAP) were also deployed to measure in situ aerosols at three visible wavelengths providing scattering coefficients and absorption coefficients, respectively."*

*Line 135: sampling off (from/by/using?) the same inlet line?*

We have rephrased this (line 135 (170)) to *"sampling from the same inlet"*.

*Line 141-142: "as the ship. . . at that time". It is not clear to me what do you mean by "restarting the system". I also feel this sentence does not contribute to the manuscript.*

This was intended to explain why the data set ended well short of port even before entering the Territorial Seas. However, it is a minor point that is not essential so we have removed that sentence and part of the preceding sentence. The conclusion of Section 2.1 now reads:

*"They also ensured that the system ran properly and were there to handle problems. The methodology and results of the filter sampling will be presented in the companion paper (Part 2, Jordan et al., 2020b)."*

*Line 148-149: the internet links are not needed and in my opinion reduce readability.*

As with the introduction of the term TAP, we have revised this text primarily in response to Referee #1's request to revise Section 2. We have also removed the internet links from lines 148-149 (180-182):

*"The TAP (model 2901, Brechtel, Hayward, CA) measures absorption coefficients ($\sigma_{abs}$) at 467, 528, and 652 nm with 1 s resolution and the IN101 (AirPhoton, Baltimore, MD) measures scattering coefficients ($\sigma_{scat}$) at 450, 532, and 632 nm with ~10 s resolution."*

*Line 165: here, the reader is asked to compare figure S3 and figure 2 but there is no explanation as to what is the difference between the two figures. It should also be made clear that this data was acquired by using the TAP and IN.*

We have updated the parenthetical in the sentence on line 165 (221-222) as follows (note the new number, S4, due to the addition of the new Fig. S2):

*"(the interested reader can compare $\sigma_{abs}$ in Fig. S4 that shows all of the data in the time series including ship plume interceptions to $\sigma_{abs}$ in the top panel of Fig. 2 where the plume interceptions have been removed)"*

*Line 174: a discussion about the errors generated from the wavelength adjustment is needed.*

It is important to remember that scattering is the dominant term in NT $\sigma_{ext}$ by about an order of magnitude, so the small wavelength adjustments made to shift $\sigma_{abs}$ from 467 to 450 nm, 528 to 532 nm, and 652 to 632 nm make a negligible contribution to the error of NT $\sigma_{ext}$.

*Line 263-265: how were local ship emissions distinguished from emissions of the research vessel?*

We have appended the following sentence to the end of that paragraph (starting at line 334 of the revised manuscript):

*"Note, the inference that aerosols during the Blocking period were from local ship emissions refers to the regional ambient environment and should not be confused with ship plume contamination from the R/V Onnuri (see Section 2.2 for the criteria used to remove ship stack contamination from the data set)."*

*Line 272: the plot referring to the 632nm channel shows that all fitted lines consistently show 5-9% over estimation of extinction from sum of scattering and absorption compared with measured extinction. What could be the cause? How accurate are the absorption measurements in this wavelength? Is it possible that these have a positive bios?*

Scattering is the dominant term here, so the contribution of absorption to the bias is likely to be small. As we discuss in Section 2.2 (lines 156-157 (212)) we were unable to perform calibrations of the nephelometer during the cruise (as is customary for this kind of measurement) and had to instead rely on pre-and post-deployment calibrations performed in our laboratory. Whether the bias you note is a fundamental characteristic of this particular IN101 or was a singular occurrence due to inadequate calibration of the nephelometer throughout the deployment is unknown. We will continue to evaluate this question with more recent data sets.

*Line 280-281: it is not clear to me what is meant here by "10 s scat and 1 s abs measurement for every ext spectrum. Please explain.*

We have revised the preceding sentence (lines 279-280 (351-353)) to clarify this:

*"This is partly attributable to the differing noise characteristics of the measurement techniques, but it also arises from differences in sampling intervals where the standard error of the means reduces the variability by the square root of the number of samples in the mean."*

*Line 303: it would be beneficial to choose one unit system (wavelength in nm or um) and b consistent throughout the manuscript.*

Referee #1 made a similar comment (major comment #4). For your convenience, we copy the response to that here:

In our previously published work (Jordan et al., 2015) and various presentations at meetings we have consistently used nm to refer to the wavelength range of the SpEx instrument. In general, it is more convenient to use integers rather than fractional values. The switch to μm in Section 3.3 arises from the need to use units consistent with those from the Schuster et al. (2006) publication in order to compare our results to theirs. In response to comments from Referees #2 and #3 we have revised the beginning of Section 4 to include more detail on our nomenclature and the implications of Eq. (5). This includes a better explanation on the necessity to be cognizant of the units when calculating spectral curvature coefficients. We think this is an important point to make, since the calculation of $\alpha_{ext}$ is wavelength independent and those who use it to extrapolate measurements from a few visible wavelengths may not be aware that using nm rather than μm for curvature calculations is problematic. For this reason, keeping both sets of units in this work is more informative.

*Line 350-355: this is an important paragraph which I strongly feel that is not explained properly. (1) "The two expressions used here to fit the relationship between σext and λ are related by their negative derivative, defined as α in Eq. (2)". Eq. 2 applies by definition only to the linear fit so how does it relate the linear fit to the 2nd order poly' fit? (2) "the derivative of the linear fit (y=a+bx; dy/dx = b) equals the derivative of the 2nd order polynomial fit (y = a0*

*+a1x+a2x2; dy/dx = a1 +2a2x) such that. . ." this statement seems wrong to me. It implies that for every λ the expression (a1+2a2\*ln(λ)) has the same value. Did you mean to write that Eq 5 is only valid for one λ? That is λch?*

We appreciate this comment and have (we hope!) done a better job of explaining what we mean in this revision. After adding a sentence immediately after Eq. (5) in response to a comment from Referee #2, we revised the remainder of the text in the paragraph on lines 354-362 (441-450) as follows:

*"For any given spectrum, there is one wavelength at which the linear and $2^{nd}$ order polynomial fits yield equivalent results in Eq. (5). This must not be confused with every wavelength measured in the spectrum, so we will refer to this one wavelength as the characteristic wavelength of the measurement range, $\lambda_{ch}$, from here on. It can be calculated for each measured spectrum from the two sets of fit coefficients for that spectrum. That is, rewriting Eq. (5) in terms of $\lambda_{ch}$ the characteristic wavelength of the measured spectrum may be calculated from $\lambda_{ch} = e^{\wedge}((\alpha_{ext} + a_1) / -2a_2)$. For the SpEx data set, $\lambda_{ch}$ was found to range from 0.36 - 0.46 µm. In contrast, the empirical fit of $\alpha_{ext} = a_2 - a_1$ implies $\lambda_{ch} \sim 0.61$, i.e., ln(0.61) ~ -0.5. The dependence of Eq. (5) on the characteristic wavelength, results in spectra sets with differing $\lambda_{ch}$ exhibiting different mapping between $\alpha_{ext}$ and $(a_1, a_2)$. To illustrate this, consider the range of $\alpha_{ext}$ values (0.29 - 3.25) found from linear fits over 0.3 - 0.7 µm to all of the spectra measured by SpEx during KORUS-OC. This range of $\alpha_{ext}$ values maps differently into $(a_1, a_2)$ space as a function of $\lambda_{ch}$ (Fig. 9, top right panel)."*

*Line 357: "The wavelength dependence of Eq. (5)" if I understand correctly, Eq. 5 is not dependent on wavelength (i.e. the spectra of the measured data, 300-700 nm). It is dependent on the characteristic wavelength (λch) as it is defined in the lines above.*

Great catch! Thanks so much, we have revised this sentence as follows (and you will also have seen it in the revised paragraph included in the response to your comment above):

*"The dependence of Eq. (5) on the characteristic wavelength, results in spectra sets with differing $\lambda_{ch}$ exhibiting different mapping between $\alpha_{ext}$ and $(a_1, a_2)$."*

*Line 360: did you mean figure 9? Not sure how figure 10 is related here.*

Yes, we did. Thanks for catching that typo on line 360 (450), it's fixed now.

*Line 360-362: it is clear that when a2 = 0 the 2nd order poly' fit is actually a line fit. What is the physical interpretation of λch = 1 um?*

This is a great question! We have added two new paragraphs (starting at line 452 of the revised manuscript) to explain the implications of $\lambda_{ch}$ and the special ("limiting" in the previous version wasn't quite the right word) cases immediately following the updated paragraph in response to your comment above regarding lines 350-355 of the submitted manuscript. We have also added Figure S8 to the supplement and include it here with the revised text for your convenience.

*"There are two special cases evident in Eq. (5) that result in $\alpha_{ext} = a_1$. First, when there is no curvature ($a_2 = 0$), $a_1$ describes the same linear fit as $\alpha_{ext}$. Second, when $\lambda_{ch} = 1$ µm (i.e., ln(1) = 0) Eq. (5) is insensitive to curvature such that $a_2$ can have any value at all. This can be understood from Eq. (1), where $\alpha$ can be any value when $\lambda = 1$ µm and $p(1µm)$ will always $= \beta$. The former leads to all $\lambda_{ch}$ sets overlapping at $a_2 = 0$, while the latter exhibits a broad vertical band independent of curvature ($a_2$) (Fig. 9, top right panel). These special cases have important implications. As $\lambda_{ch}$ approaches 1 µm the measurement becomes insensitive to curvature, while at the short wavelengths of light represented by $\lambda_{ch} < 0.1$ µm the curvature itself becomes unimportant. Hence, to probe spectral curvature the upper right panel of Fig. 9 shows measurement techniques with $\lambda_{ch} \sim 0.5 \pm 0.2$ µm provide the greatest sensitivity with sufficient separation in ($a_1,a_2$) to distinguish aerosol microphysical and chemical properties influencing the spectral shape.*

*The rotation as a function of $\lambda_{ch}$ shown in Fig. 9 also illustrates why wavelength units of µm must be used to calculate ($a_1,a_2$). As $\lambda_{ch}$ increases to values > 1 µm, the $\alpha$ map rotates clockwise (Fig. S8). If one used $\lambda_{ch} = 410$ nm rather than 0.41 µm, it would map into a narrow band in the next quadrant of ($a_1,a_2$) space spanning a wide range in $a_1$ but a narrow range in $a_2$, resulting in little curvature sensitivity. The calculation of $\alpha_{ext}$ is wavelength independent and will produce the same result no matter what units are used. This is not the case for the calculation of ($a_1,a_2$), so it must be emphasized that for curvature, the units matter."*

[Figure]

*"**Figure S8.** Illustration of the rotation of the mapping of $\alpha$ in ($a_1,a_2$) space as a function of $\lambda_{ch}$ for long wavelengths. This illustration includes the extreme values of $\lambda_{ch}$ of 100 and 1000 µm to stand in for the mapping that arises from using nm units instead of µm of wavelength."*

Line 364: what does it mean?

This is now better explained in the new paragraphs provided in the response to the previous comment.

*Line 369: to the right and left of which distribution?*

We have revised the sentence on line 369 (502) to read:

*"Note, that the most extreme values in $a_1$ for the KORUS-OC data set are related to shorter $\lambda_{ch}$ than the rest of the data set."*

*Lines 376-378: "but as discussed. . .space". This section is realy not clear. If the intension here was to link the information about the sampled aerosols that is available in other publications and that is presented in section 3.1 to the fitted parameter space, this is was not made clear. Please rephrase.*

We have revised the text on lines 376-378 (508-511) as follows:

*"Aerosol size distributions were not measured aboard ship during the cruise but as described in Sect. 3.1, previously published work provides sufficient information for a broad characterization of the different ambient aerosol populations prevalent during the three meteorological regimes that occurred during KORUS-OC. This context is used to assess the mapping of SpEx data into $(a_1, a_2)$ space."*

*Figure 4 top: the gray trace is not visible.*

As explained in the figure caption:

*"Top panel: SpEx (all data, gray; above LLOD, green; **these curves are coincident until June 2$^{nd}$ when the lowest values are below detection and hence, appear gray**)"*

There are very few spectra for which 532 nm $\sigma_{ext}$ is below the LLOD, but for the smallest values in the time series after June 2$^{nd}$, the reader will note that there are a few gray points visible below the majority of data in the time series in green.

*Figure 6: it would be more intuitive for me to plot these data sets with the EAE calculated from the full range on x axis because this represents the optimal case and demonstrates the variances is most of the spectra to that case.*

The choice of using 450-632 nm for the wavelength range on the x-axis is primarily intended for readers accustomed to $\alpha_{ext}$ values calculated from a few visible wavelengths from the sum of $\sigma_{scat}$ and $\sigma_{abs}$ measured with standard commercial instruments (e.g., similar to the NT $\sigma_{ext}$ and NT $\alpha_{ext}$ in this work). Compared to values typically obtained from visible only wavelength ranges, $\alpha_{ext}$ values from the full SpEx wavelength range are smaller and this is the primary difference to which we want to draw the reader's attention.

*Figure 7: what information is presented or made clearer in the top two figures that is not presented or is not clear in the middle two figures? In my opinion there is no add value to the top two and therefore should be removed. Additionally, the residual could be presented in relative terms (i.e. % error). This would be more intuitive to understand and reduce the text needed to describe the fit errors.*

We feel that the top two panels of Fig. 7 are essential for providing context for the middle panels as discussed in the last 3 paragraphs of Section 3 starting at line 313 (398). Figure 7 is primarily for the benefit of those who by necessity extrapolate $\sigma_{ext}$ into the UV from visible in situ measurements using $\alpha_{ext}$ to alert them to the potential for large errors. Further, Fig. 7 shows that extrapolating from 2$^{nd}$ order polynomials doesn't resolve this problem and therefore as the final sentence of Section 3 states:

*"The divergence of either fit from the measured spectrum when extrapolating beyond the fit wavelength range (Fig. 7, middle panels) highlights the need for measurements across a broad spectral range in order to minimize the need for extrapolation."*

Additionally, we use the residuals in this figure primarily to illustrate the improvement in the fit (i.e., the reduction in curvature in the top and bottom pair of panels) between the linear and 2$^{nd}$ order polynomials. Keeping the units the same between the main panel with the red spectrum and the black curve fit and the upper panel with the residuals is necessary to make this point.

*Figure 8: here I would also suggest showing the x-axis in terms of relative error.*

This figure is similar to one in Part 2 and both are intended to provide a means to illustrate that the improvement in the residuals from linear to 2$^{nd}$ order polynomial fits for the specific example spectra shown here in Fig. 7 here and in Fig. S4 of Part 2 apply to the data set as a whole. For this reason, we would prefer to keep the focus on the residuals rather than relative error in these figures.

*Figure 9: in this figure all x-axis are inverted. This makes the figure less intuitive and harder to understand. Is there a reason for this choice? If yes I believe it should be explain in the text or at least in the figure caption. What is the purpose of the rectangle inset in the bottom panel of figure 9?*

As discussed above the x-axes follow the previously published work of Schuster et al. (2006) to which we compare our results. We have updated the figure caption for Fig. 9 to conclude with the following explanation for the black box in the bottom panel:

*"The black box is the same as the one in the top left panel."*

*Figure S2: it is not clear what this data set is based on. Is it simulation or measurements? If data is from another reference which is it?*

This is a calculation. We have updated the figure caption as follows (note that this in now in reference to Fig. S3 due to the addition of the new Fig. S2):

**"*Figure S3.  Illustration of the relationship between flow rate and size cut of the sampled aerosol particles based on theoretical calculations pertinent to the system deployed aboard the R/V Onnuri.*"**

*Figure S3: the full range of the right axis is not needed. Values range from 0 to 2.*

The scale of the right axis was chosen to ensure that the black trace of the plume flag values did not interfere with the $\sigma_{scat}$ and $\sigma_{abs}$ curves above it.

*Figure S4: what do the authors mean by hour set of intensity spectra? Is it a one hour average?*

We have updated the figure caption as follows (again, note the updated figure number due to the new Fig. S2):

**"*Figure S5.  Example of a one hour set of intensity spectra that illustrate the two wavelengths (332 nm and 467 nm) that sometimes saturated due to drift in the lamp intensity.  30 individual spectra measured at 2 min intervals over the course of one hour are shown.  Typically (as is the case here) there is little discernible difference in intensity between sample and reference spectra over the full range of counts in the 16 bit spectrometer (0-65,536 counts).*"**